# Whole exome sequencing analysis identifies genes for alcohol consumption

Jujiao Kang [1,2,9], Yue-Ting Deng [3,9], Bang-Sheng Wu[3,9], Wei-Shi Liu[3], Ze-Yu Li[1,2], Shitong Xiang [1,2], Liu Yang[3], Jia You [1,2], Xiaohong Gong [4], Tianye Jia [1,2,5,6], Jin-Tai Yu [3] ✉, Wei Cheng [1,2,3,7] ✉ & Jianfeng Feng [1,2,7,8] ✉

Alcohol consumption is a heritable behavior seriously endangers human health. However, genetic studies on alcohol consumption primarily focuses on common variants, while insights from rare coding variants are lacking. Here we leverage whole exome sequencing data across 304,119 white British individuals from UK Biobank to identify protein-coding variants associated with alcohol consumption. Twenty-five variants are associated with alcohol consumption through single variant analysis and thirteen genes through gene-based analysis, ten of which have not been reported previously. Notably, the two unreported alcohol consumption-related genes *GIGYF1* and *ANKRD12* show enrichment in brain function-related pathways including glial cell differentiation and are strongly expressed in the cerebellum. Phenome-wide association analyses reveal that alcohol consumption-related genes are associated with brain white matter integrity and risk of digestive and neuropsychiatric diseases. In summary, this study enhances the comprehension of the genetic architecture of alcohol consumption and implies biological mechanisms underlying alcohol-related adverse outcomes.

Alcohol consumption is a prominent risk factors for death and disability worldwide, accounting for over two million deaths each year[1]. It poses a tremendous threat to human health through multiple mechanisms, including cumulative damage to organs and leading to self-harm or violence[2,3]. Notably, these adverse effects are largely dependent on the average volume of alcohol consumption[4]. Identifying the risk factors that influence one's level of alcohol consumption can contribute to the prevention, identification, and treatment of adverse outcomes from alcohol consumption[5].

Over the recent decades, comprehensive genome-wide association studies (GWAS) have indicated the potential influence of genetic

factors on one's alcohol consumption volume and identified over 100 related variants[6,7]. However, a predominant proportion of the identified variants are localized within noncoding regions, and their effect sizes tend to be small, making interpretation and identification of the causal gene challenging[8]. In addition, previous GWAS mainly utilized imputed genotype data, which only cover limited regions of the genome, and thus may have missed many potential genes. Furthermore, GWAS studies focused mainly on common variants, and few studies have investigated rare variants associated with alcohol consumption, which yield greater potential to interpret biological function and elucidate mechanisms[9]. Although there are studies that have attempted

[1]Institute of Science and Technology for Brain-Inspired Intelligence (ISTBI), Fudan University, Shanghai 200433, China. [2]Key Laboratory of Computational Neuroscience and Brain-Inspired Intelligence (Fudan University), Ministry of Education, Shanghai 200433, China. [3]Department of Neurology and National Center for Neurological Disorders, Huashan Hospital, State Key Laboratory of Medical Neurobiology and MOE Frontiers Center for Brain Science, Shanghai Medical College, Fudan University, Shanghai 200433, China. [4]School of Life Sciences, Fudan University, Shanghai 200433, China. [5]Social Genetic and Developmental Psychiatry Centre, Institute of Psychiatry, Psychology and Neuroscience, King's College London, London, UK. [6]School of Psychology, University of Southampton, Southampton, UK. [7]Fudan ISTBI—ZJNU Algorithm Centre for Brain-inspired Intelligence, Zhejiang Normal University, Zhejiang, China. [8]Department of Computer Science, University of Warwick, Coventry CV4 7AL, UK. [9]These authors contributed equally: Jujiao Kang, Yue-Ting Deng, Bang-Sheng Wu. ✉e-mail: jintai_yu@fudan.edu.cn; wcheng@fudan.edu.cn; jffeng@fudan.edu.cn

to leverage exome chip data to identify rare variants contributing to alcohol consumption, the sample size was small and limited regions of the whole exome were examined[10].

The introduction of whole exome sequencing (WES) provides a great chance to overcome the limitations of previous genetic studies on alcohol consumption with a substantially larger amount of rare and ultra-rare protein-coding variants[11–13]. Collapsing of loss-of-function (LOF) variants helps estimate the effect direction of associated genes[13,14]. When combined with large-scale population cohorts with multi-modal phenotypic data, WES would greatly facilitate our understanding of the genetic underpinnings of alcohol consumption as well as its implication on physical and mental health[6]. However, to our knowledge, there have been few large-scale WES studies on alcohol consumption, let alone elucidating the potential implications of the identified genes[10,15]. Meanwhile, as indicated by a previous genome-wide association study, significant genetic associations existed between alcohol consumption and several body health phenotypes[7]. The application of phenome-wide analysis for alcohol-related genes can help extend and deepen our current comprehension of the association between alcohol consumption and human health.

Hence, aiming to refine the genetic architecture of alcohol consumption, we conduct an exome-wide association study (ExWAS) for alcohol consumption among 304,119 individuals from the UK Biobank (UKB). We also examine the rare-variant associations with genes reported by previous GWAS[6,7,16,17]. Finally, we provide biological insights into the identified genes via bioinformatics analyses and phenome-wide association analysis (PheWAS).

## Results

### Study population and data description

We leveraged exome sequencing data and phenotypic data from UKB and excluded low-quality variants and samples (Methods)[13,18]. For the main analysis, we included 304,119 unrelated white British participants. The average age was 56.87 years at enrollment and 54.09% participants were female. Information about alcohol drinking per week were obtained from self-completed touchscreen interviews at baseline (Methods and Supplementary Data 1). The average alcohol consumption (alcohol amounts after natural logarithm) of the whole sample was 2.06 (Standard Deviation (SD) = 1.44), with a mean of 2.47 (SD = 1.41) and 1.72 (SD = 1.38) for males and females respectively (Supplementary Data 2). Finally, the exome-wide association analysis included 100,101 common variants (with a MAF of ≥1%) and 13,018,630 rare variants (with a MAF of <1%). Figure 1 provided the general schema of our study.

### ExWAS for alcohol consumption

To test whether alcohol consumption was associated with damaging coding variants, we conducted ExWAS using a linear mixed model with adjustments for ten principal components, age, and sex (Methods). The analysis discovered two rare variants and 23 independent common variants linked to alcohol consumption ($P < 5 \times 10^{-8}$) (Table 1, Fig. 2a, b). The genomic control lambda is 1.04, indicating that the association statistics are not systematically inflated (see Supplementary Fig. 1 for the corresponding quantile-quantile plot). The top rare variant, rs283413 (MAF = 0.8%; $\beta_A = -0.15$, $P = 2.73 \times 10^{-31}$) is a stop-gain variant in *ADH1C*, the well-known gene related to alcohol metabolism. Among the 23 common variants, three were not reported previously (rs41288799, rs4975020 and rs77623289). Most of the identified variants are intron (46%) or missense (19%) (Fig. 2c, Supplementary Data 3, Methods). Additionally, 15 of the 22 identified variants, which were examined in an independent alcohol consumption GWAS[19], showed nominal significance ($P < 0.05$) (Table 1, Supplementary Data 4). Further, 17 of the 24 identified variants available in the FinnGen study[20] exhibited nominal associations with alcohol use disorder (AUD)

($P < 0.05$) (Supplementary Data 5). To assess the robustness of the main analysis, we adjusted for rs1229984, a well-established marker strongly linked to alcohol consumption[6,21]. Notably, 23 of 25 variants (92%) retained the same association directions, with 22 variants (88%) maintaining their significance ($P < 5 \times 10^{-8}$) (Supplementary Data 6). Additionally, the main analysis maintained its robustness after excluding former drinkers and non-drinkers. Further, all effect directions remained the same, and 20 of the initially identified 25 variants (80%) retained their significance ($P < 5 \times 10^{-8}$) (Supplementary Data 7). Finally, the ExWAS for scores of alcohol use disorders identification test (AUDIT) identified a rare variant (rs283413) and two independent common variants (rs13107325 and rs201168482) associated with alcohol use problems (Supplementary Figs. 2–4, Supplementary Data 8).

Since a single rare variant tends to be of insufficient power to identify significant signals, we further performed gene-based collapsing analysis to detect genes related to alcohol consumption. LOF and missense rare variants of each gene and three MAF thresholds (<1%, <0.1% and <0.01%) were utilized. In total, we identified 19 associations (covering seven genes) after Bonferroni correction (Table 2 and Fig. 3a; Supplementary Data 9; $P < 0.05/19852 = 2.5 \times 10^{-6}$). Rare variants in the known alcohol consumption-related gene, *ADH1C* showed the most significant gene-based association at $P = 1.91 \times 10^{-30}$. The maximum genomic control lambda was 1.076 (see Supplementary Fig. 5 for the corresponding quantile-quantile plots). The total rare burden heritability of alcohol consumption was 0.88% (Fig. 3b and Supplementary Data 10). We additionally identified six putative alcohol consumption-related genes under the threshold of overall false discovery rate (FDR) < 0.05 ($P < 1.69 \times 10^{-5}$). Among these rare-variant genes, seven (*GIGYF1*, *ANKRD12*, *KDM5B*, *APC2*, *LGI2*, *ATP1A2*, and *ENSG00000224076* (not officially designated and excluded from further analysis)) were not previously reported in GWAS studies for alcohol consumption. The LOF and missense burden in eleven of the rare-variant genes reduced alcohol consumption ($\beta = -0.003$ to $-0.023$; Fig. 3c, Table 2). In addition, 2.03% ($n = 8825$) of the participants carried a LOF variant located in *ADH1C* exons and *GIGYF1* variants were carried by 1.72% ($n = 7449$) participants (Fig. 3d). After excluding the former drinkers and non-drinkers, 19 out of the initially identified 39 associations retained the significance ($P < 1.69 \times 10^{-5}$, Supplementary Data 11). Following adjustment for rs1229984, the identified associations were robust except for *ADH1C*, *ADH1A*, *SNX17* and *ADH5* (Supplementary Data 12). Additionally, we performed ExWAS for AUDIT and identified two genes (*ADH1C* and *CA1*) associated with alcohol use problems (Supplementary Figs. 6-8, Supplementary Data 13).

### Leave-one-variant-out (LOVO) and conditional analysis

To investigate whether a single variant dominated the gene-based associations, we firstly conducted LOVO analysis. While the maximum $P$-value for *ADH1C* was $P = 0.802$ after the removal of rs283413, $P = 0.873$ for *ADH1A* after the removal of rs190428650, $P = 0.446$ for *SNX17* after the removal of rs147740391, and $P = 0.016$ for *ADH5* after the removal of rs62325244, the other associations did not exhibit substantial attenuation (Supplementary Figs. 9–20, Supplementary Data 14). Hence, even a single variant, i.e. of *ADH1C*, *ADH1A*, and *ADH5*, may critically influence alcohol consumption, whereas the other significant associations were based on a burden of multiple rare variants. Subsequently, conditional analysis was performed to assess whether the significant associations with rare variants were influenced by adjacent common variants (Methods). Seven genes were found to have nearby common variants exhibiting significant associations with alcohol consumption. The associations of *GIGYF1*, *ANKRD12* and *APC2* did not exhibit substantial attenuation, whereas the associations of *ADH1C*, *ADH5* and *SNX17* exhibited attenuation, though still nominally significant, and the association of *ADH1A* lost its significance after adjustment for the nearby common variants (Supplementary Data 15).

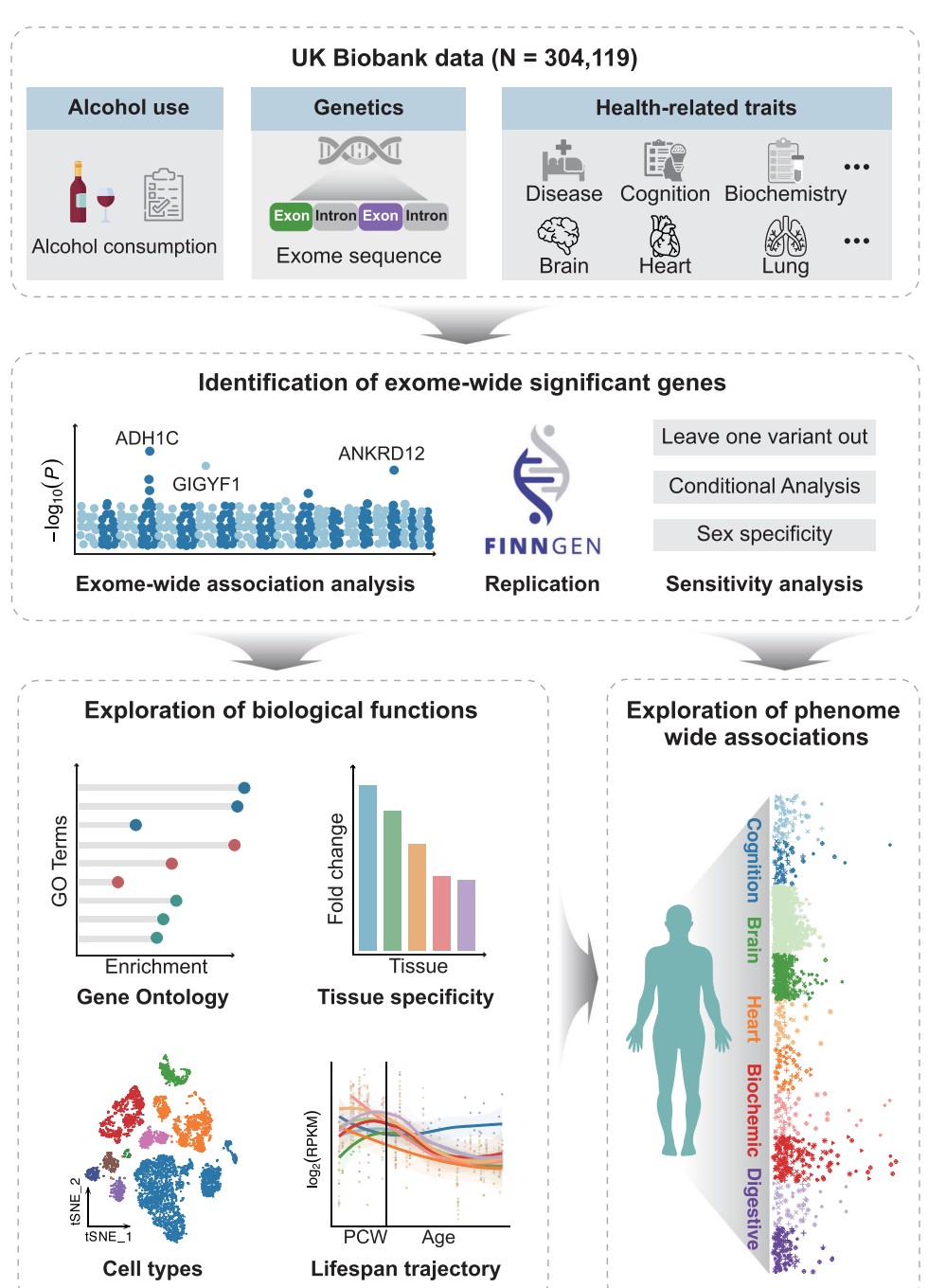

**Fig. 1 | Study overview.** The top section outlines the data utilized in the study, including alcohol use, exome sequence data, and health-related phenotypes. The middle section outlines the identification of exome-wide significant genes, involving exome-wide association analysis, replication in the FinnGen cohort, and sensitivity analysis. The bottom-left section outlines biological functions analysis of the identified genes, including GO analysis, tissue expression enrichment analysis, cell-type expression, and lifespan spatio-temporal brain expression trajectory analysis. The bottom-right part focuses on exploring phenome-wide associations of the identified genes. GO Gene Ontology, PCW Post-conception weeks, RPKM Reads per kilobase million; tSNE t−Stochastic Neighbourhood Embedding. 'image: Flaticon.com'. This cover has been designed using images from Flaticon.com.

## Sex-specific analysis of the associations

As the average alcohol consumption showed a significant difference between males and females, we conducted gene-based collapsing analyses on participants separated by sex to explore whether the genetic contributions to alcohol consumption also differed by sex. While the *KDM5B* gene's association with alcohol consumption was only observed in males ($P = 3.04 \times 10^{-7}$ for males and $P = 0.170$ for females), the other genes were significantly associated with alcohol consumption in both males and females ($P < 0.05$, Supplementary Data 16).

## Associations of rare variants in alcohol-related genes

We then examined the impact of rare variants based on previous GWAS findings on alcohol consumption. We assessed a total of 174 alcohol consumption-related genes identified by the most recent GWAS studies[6,7,16,17]. Although 25 genes showed nominal significance, only the *ADH1C* gene was significant after Bonferroni correction (Supplementary Data 17). The influence of coding variants within the GWAS regions did not exhibit substantial effects, potentially due to the limited statistical power of ExWAS.

**Table 1 | Exome-wide significant variants for alcohol consumption**

| CHR | SNP | A1 | A2 | *SYMBOL* | AF | β | SE | P |
|-----|-----|----|----|----------|-----|-----|-----|-----|
| **2** | **rs41288799** | G | C | ***PREB, ABHD1*** | 0.037 | −0.03 | 0.006 | 3.23E-08 |
| 2 | rs3214499 | G | GA | *NRBP1* | 0.447 | 0.02 | 0.002 | 1.28E-22 |
| 2 | rs3811644 | A | G | *C2orf16* | 0.210 | 0.02 | 0.003 | 3.94E-09 |
| 4 | rs149109767 | AGAG | A | *HTT* | 0.074 | −0.02 | 0.004 | 4.07E-08 |
| 4 | rs11096989 | A | AG | *WDR19* | 0.437 | 0.01 | 0.002 | 3.78E-09 |
| 4 | rs4975015 | T | C | *KLB* | 0.186 | 0.02 | 0.003 | 8.61E-10 |
| 4 | rs4975017 | C | A | *KLB* | 0.328 | −0.02 | 0.002 | 7.20E-12 |
| **4** | **rs4975020** | A | G | ***UGDH*** | 0.324 | 0.01 | 0.002 | 4.08E-09 |
| 4 | rs190428650 | C | G | *ADH1A* | 0.000 | −0.29 | 0.052 | 2.53E-08 |
| 4 | rs283413 | C | A | *ADH1C* | 0.008 | −0.15 | 0.013 | 2.73E-31 |
| 4 | rs17526590 | G | A | *ADH1C* | 0.096 | 0.03 | 0.004 | 3.94E-15 |
| 4 | rs113337987 | G | A | *MTTP* | 0.020 | −0.06 | 0.008 | 4.70E−13 |
| 4 | rs13107325 | C | T | *SLC39A8* | 0.074 | −0.04 | 0.004 | 1.30E-20 |
| 7 | rs13235543 | C | T | *MLXIPL* | 0.129 | 0.02 | 0.003 | 5.26E-09 |
| 11 | rs755555 | C | T | *SLC39A13* | 0.320 | −0.02 | 0.002 | 7.94E-12 |
| 11 | rs10891540 | A | G | *TTC12* | 0.463 | 0.01 | 0.002 | 3.31E-08 |
| 11 | rs1800497 | G | A | *ANKK1* | 0.201 | 0.02 | 0.003 | 4.92E-08 |
| 12 | rs2400895 | A | T | *ACSS3* | 0.450 | 0.01 | 0.002 | 4.70E-08 |
| 14 | rs28929474 | C | T | *SERPINA1* | 0.021 | −0.05 | 0.008 | 3.07E-09 |
| **15** | **rs77623289** | G | T | ***ISL2*** | 0.053 | 0.03 | 0.005 | 1.81E-08 |
| 16 | rs62036622 | T | G | *ATXN2L* | 0.420 | −0.01 | 0.002 | 7.26E-10 |
| 16 | rs2278557 | C | G | *PPP4C* | 0.404 | −0.01 | 0.002 | 2.72E-08 |
| 17 | rs8073146 | A | G | *CRHR1* | 0.226 | −0.02 | 0.003 | 3.83E-15 |
| 18 | rs1788825 | C | T | *RMC1* | 0.346 | −0.01 | 0.002 | 3.80E-10 |
| 19 | rs516246 | T | C | *FUT2* | 0.492 | −0.02 | 0.002 | 3.77E-12 |

SAIGE GENE+ was utilized to conduct single-variant association tests (two-sided). Independent significant ($P < 5 \times 10^{-8}$) variants for alcohol consumption are presented. No corrections were applied for multiple comparisons. A variant is reported if the mapped genes have been previously reported for alcohol use according to the GWAS catalog, with boldface indicating associations not previously identified. *CHR* Chromosome, *A1* Allele 1; *A2* allele 2, *AF* Allele frequency of allele 2, *β* β value for allele 2.

## Biological function and tissue expression of the alcohol consumption-related genes

We further conducted a series of bioinformatics analyses to investigate the biological functions of the alcohol consumption-related genes. We first performed pathway enrichment analyses. We found the enrichment of gene ontology (GO) pathways relevant to alcohol dehydrogenase activity, oxidoreductase activity, ethanol oxidation and ethanol metabolism (Fig. 4a, Supplementary Data 18). Also, the analysis of Kyoto Encyclopedia of Genes and Genomes (KEGG) pathways identified the enrichment of these genes in tyrosine metabolism, fatty acid degradation, and pyruvate metabolism. These results hence supported the biological validity of our genetic findings.

We further analyzed tissue-specific expression enrichment of the identified genes based on the Human Protein Atlas project using the TissueEnrich R package[22]. We observed six, four, and two genes enriched in the liver, duodenum, and adipose tissue, respectively (Fig. 4b, Supplementary Fig. 21, and Supplementary Data 19). Genes, including *SERPINA1*, *ADH1C*, *ADH1A*, *MLXIPL*, *MTTP*, and *KLB* were specifically enriched in the liver (Supplementary Fig. 22). Subsequently, we evaluated the expression levels of these six genes across various cell types in the liver with single-cell RNA sequencing (scRNA-seq) data. While *SERPINA1* was widely expressed in all cell types, *ADH1A*, *ADH1C*, *MTTP*, and *MLXIPL* were all predominantly expressed in the hepatocytes (Fig. 4c, d).

We further estimated the similarities between genes based on the association results of collapsing analyses across 1419 quantitative traits in UKB using Gene-SCOUT[23]. Notably, the *GIGYF1* gene exhibited the highest similarity to *ANKRD12* (Fig. 5a and Supplementary Data 20). Interestingly, the top 10 similar genes of *ANKRD12* are enriched in brain function-related pathways containing glial cell differentiation,

cognitive function, and glutamate secretion (Fig. 5b and Supplementary Data 21). Thus, to gain more insights into how these rare-variant genes may be related to alcohol use, we further examined the expression of *ANKRD12* and *GIGYF1* across tissues within the Human Protein Atlas[24]. Notably, both *ANKRD12* and *GIGYF1* exhibited strong expression in the brain, particularly in the cerebellum (Fig. 5c, Supplementary Fig. 23). In addition, both *ANKRD12* and *GIGYF1* showed broad expression in all cell types in brain (Supplementary Fig. 24). We subsequently characterized the spatiotemporal expression trajectories of *ANKRD12* and *GIGYF1* in the human brain, using mRNA sequencing (mRNA-seq) data from the PsychEncode study[25]. Our findings revealed unique temporal expression patterns of these genes in the cerebellum compared to other regions of the brain (Fig. 5d, e). These results imply that these two genes associated with alcohol consumption may alter the function of brain, which are important targeted organ of alcohol intake, providing clues for future research on the alcohol-related brain injury.

## Phenotypic associations with alcohol consumption-related genes

Alcohol consumption has been documented to correlate with various biological markers, including metabolites, and health outcomes[7,26–28]. To systematically assess the relationship between genetic variation in alcohol consumption and a broad spectrum of health phenotypes, we performed PheWAS for the identified alcohol consumption-related genes across blood indices, major diseases, body function, and brain structures from the UKB (Methods and Supplementary Data 22).

Among the 82 significant gene-phenotype and 380 variant-phenotype associations ($P < 0.05/316/12 = 1.32 \times 10^{-5}$, $P < 0.05/316/25 = 6.33 \times 10^{-6}$, respectively), 81.7% and 47.4% were related to

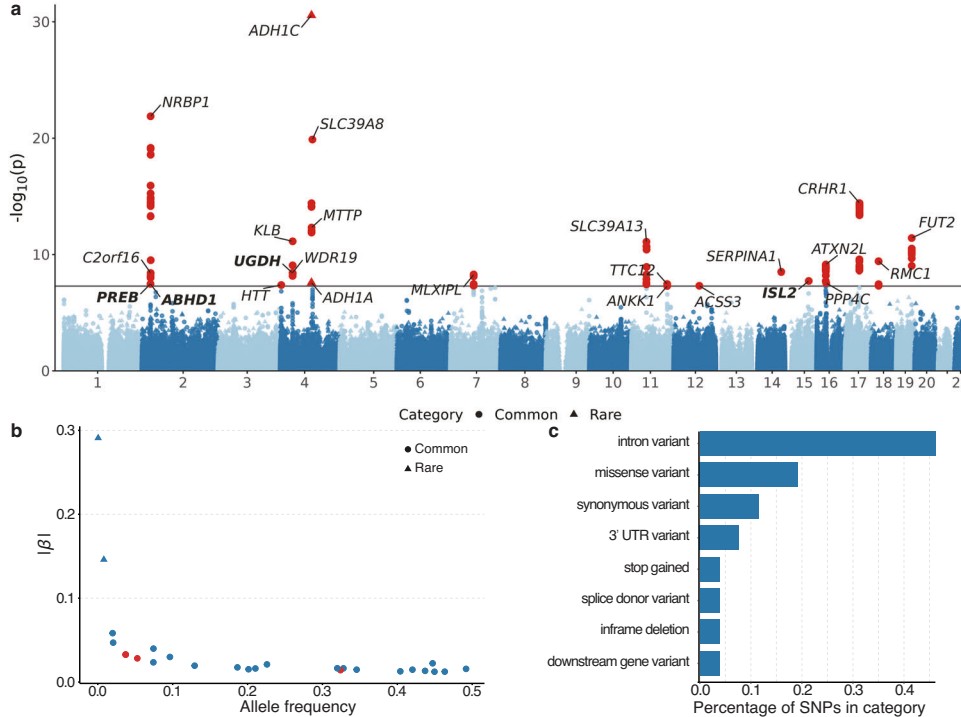

**Fig. 2 | Single-variant ExWAS of alcohol consumption. a** Manhattan plot showing the results of the common variants from ExWAS of alcohol consumption. SAIGE GENE+ was used to perform single-variant association tests ($N = 304,119$ biologically independent samples). The chromosomal position of the variant across the 22 chromosomes is represented on the x-axis, while the y-axis displays the -log$_{10}$-transformed p-value. The significance threshold ($P < 5 \times 10^{-8}$) is denoted by the horizontal black line. Models are corrected for the top ten ancestral principal components, age, and sex. The presented p-values are two-sided and have not been adjusted for multiple testing. Significant variants were marked with red. Independent significant variants were marked with the nearby genes. Genes identified in this study were marked in bold. **b** Plot of effect size (absolute value) versus allele frequency of 22 previously reported alcohol consumption variants (blue) and 3 previously not reported alcohol consumption variants (red). **c** Distribution of the functional consequences of independent significant variants.

## Table 2 | Gene associated with alcohol consumption at FDR < 0.05

| Region | Group | Max MAF | MAC | N rare | N ultra-rare | β$_{Burden}$ | P |
|---|---|---|---|---|---|---|---|
| *ADH1C* | lof | 0.01 | 4986 | 2 | 19 | −0.007 | 1.91E-30 |
| **GIGYF1** | lof | 1.00E-04 | 103 | 1 | 58 | −0.023 | 6.92E-11 |
| **ANKRD12** | lof | 1.00E-04 | 214 | 4 | 81 | −0.015 | 3.91E-10 |
| *ADH1A* | lof | 0.001 | 424 | 5 | 19 | −0.008 | 1.39E-08 |
| *ADH1A* | missense; lof | 0.001 | 935 | 19 | 68 | −0.003 | 4.05E-08 |
| **KDM5B** | missense; lof | 0.001 | 1386 | 19 | 427 | −0.004 | 6.07E-07 |
| *CTNNA2* | missense; lof | 1.00E-04 | 530 | 10 | 175 | −0.007 | 1.16E-06 |
| **APC2** | missense; lof | 0.001 | 2057 | 36 | 331 | −0.004 | 1.58E-06 |
| **KDM5B** | missense; lof | 1.00E-04 | 1124 | 16 | 427 | −0.004 | 2.26E-06 |
| **LGI2** | missense; lof | 0.001 | 621 | 11 | 117 | −0.007 | 3.48E-06 |
| **ANKRD12** | missense; lof | 1.00E-04 | 1191 | 27 | 337 | −0.005 | 3.52E-06 |
| **APC2** | missense; lof | 1.00E-04 | 1462 | 30 | 331 | −0.004 | 3.61E-06 |
| **GIGYF1** | missense; lof | 0.001 | 1310 | 29 | 198 | −0.004 | 8.72E-06 |
| **ANKRD12** | missense; lof | 0.001 | 2121 | 33 | 337 | −0.003 | 1.02E-05 |
| **ENSG00000224076** | lof | 1.00E-04 | 5 | 0 | 1 | 0.069 | 1.14E-05 |
| **ATP1A2** | missense; lof | 1.00E-04 | 512 | 9 | 166 | −0.006 | 1.29E-05 |
| *SNX17* | missense; lof | 0.01 | 5405 | 8 | 86 | 0.002 | 1.54E-05 |
| *HECTD4* | lof | 1.00E-04 | 117 | 2 | 66 | −0.012 | 1.56E-05 |
| **ATP1A2** | missense; lof | 0.001 | 574 | 10 | 166 | −0.005 | 1.63E-05 |
| *ADH5* | missense; lof | 0.001 | 1215 | 17 | 88 | −0.004 | 1.64E-05 |

SAIGE GENE+ was utilized to conduct gene-based collapsing tests. Genes for alcohol consumption with FDR < 0.05, which is equivalent to $P < 1.69 \times 10^{-5}$ are presented. No corrections were applied for multiple comparisons. Genes not previously reported to be associated with alcohol use according to the GWAS catalog were highlighted in boldface. *Region* gene name, *Group* Annotation mask, *lof* loss of function, *Max MAF* Maximum MAF cutoff, *P* p-value for SKAT-O test, *BETA_Burden* effect size of burden test, *SE_Burden* standard error of BETA_Burden.

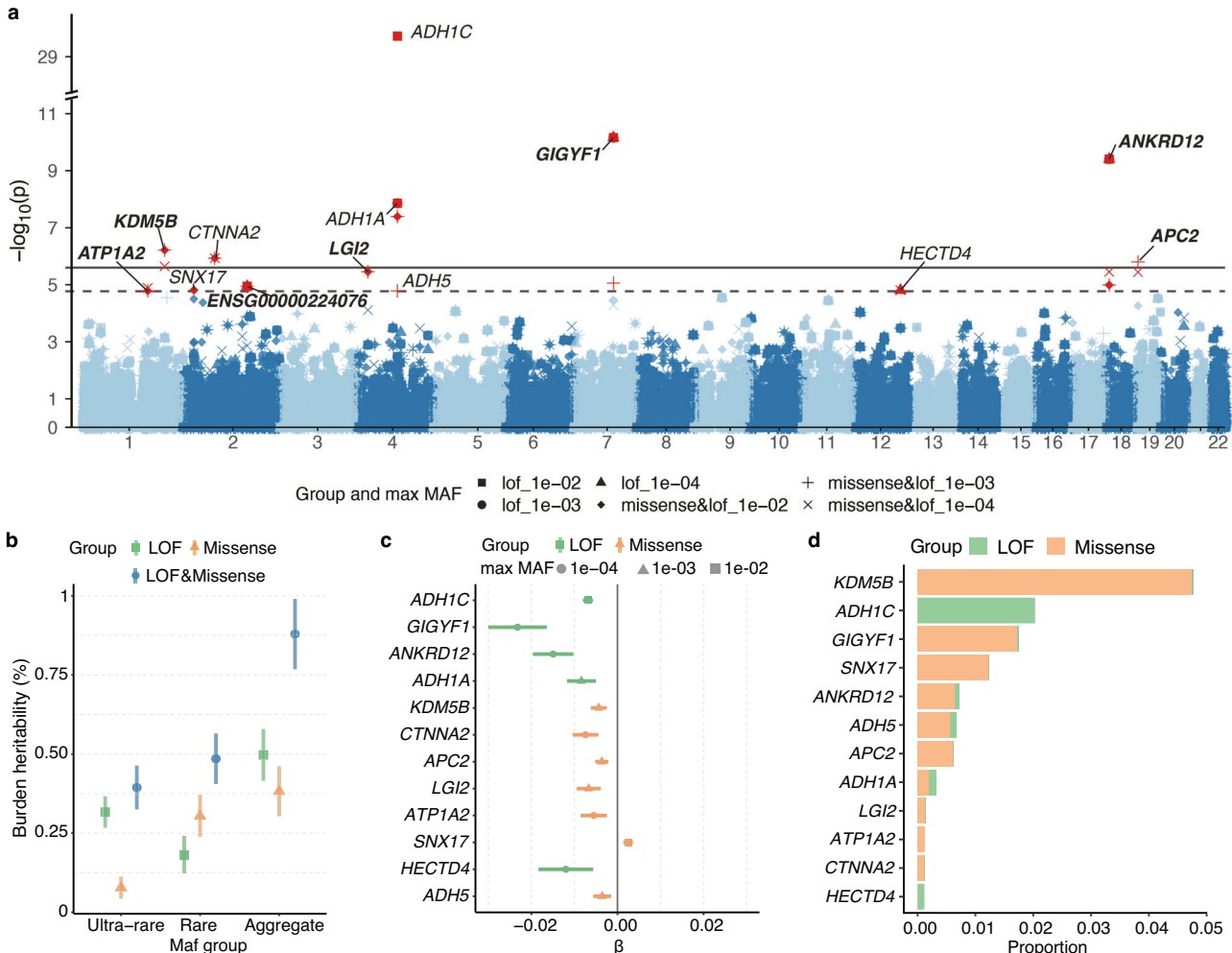

**Fig. 3 | Gene-based ExWAS of alcohol consumption. a** Manhattan plot showing the results of the rare variants (LOF and Missense) from ExWAS of alcohol consumption with three different MAF thresholds in gene-based analysis. The *x*-axis represents the gene position on 22 chromosomes. The *y*-axis indicates the -log$_{10}$-transformed p-value. Shape indicates combinations of different MAF groups and consequence groups, including LOF and Missense. The black horizontal line denotes the exome-wide significance level using Bonferroni correction ($P < 2.5 \times 10^{-6}$). The black dashed horizontal line indicates significant associations at an overall FDR < 0.05($P < 1.69 \times 10^{-5}$). Models are corrected for the top ten ancestral principal components, age, and sex. *P*-values are two-sided and unadjusted for multiple testing. Significant genes were marked with red. **b** Burden heritability of alcohol consumption in different groups. The *x*-axis denotes the different MAF groups (ultra-rare (MAF < $1 \times 10^{-5}$) and rare ($1 \times 10^{-5} \le$ MAF < $1 \times 10^{-2}$)). The color shows the LOF, Missense, and aggregation of the two groups. The *y*-axis indicates the burden heritability (h$^2$) in percent. Error bars indicate the standard error. **c** Plot of the effect sizes of burden test of the significant associations. *N* = 304,119 biologically independent samples were utilized in the analysis. For each gene, the most significant associations were plotted. Data presentation is in the form of $\beta \pm$ s.e. $\times 1.96$. **d** The carrier percentage for rare LOF and missense variants in the alcohol consumption-related genes. The color of the bar indicates LOF (green) or missense (orange) groups of the variants.

inflammatory and blood biochemistry indices (Fig. 6 and Supplementary Data 23, 24, Supplementary Figs. 25–61). Indicators of inflammation and disturbance of lipid metabolism showed significant associations with alcohol consumption-related genes. *GIGYF1* and *ANKRD12* showed the most phenotypic associations. *GIGYF1* showed strong positive associations with HbA1c ($\beta_{burden} = 0.029$, $P = 2.51 \times 10^{-13}$) and glucose ($\beta_{burden} = 0.027$, $P = 1.92 \times 10^{-10}$), and negative associations with total cholesterol level ($\beta_{burden} = -0.029$, $P = 4.52 \times 10^{-13}$), low-density lipoprotein cholesterol level (LDLC) ($\beta_{burden} = -0.026$, $P = 1.33 \times 10^{-10}$) and Apolipoprotein B ($\beta_{burden} = -0.024$, $P = 6.15 \times 10^{-10}$). *ANKRD12* showed strong positive associations with neutrophil percentage ($\beta_{burden} = 0.021$, $P = 1.19 \times 10^{-13}$) and neutrophil-lymphocyte ratio ($\beta_{burden} = 0.020$, $P = 7.34 \times 10^{-13}$), and negative associations with lymphocyte percentage ($\beta_{burden} = -0.021$, $P = 2.75 \times 10^{-13}$), total protein level ($\beta_{burden} = -0.019$, $P = 1.36 \times 10^{-10}$), and monocyte percentage ($\beta_{burden} = -0.015$, $P = 3.04 \times 10^{-8}$).

Interestingly, the gene-phenotype associations also extended to cognitive function and white matter. *ANKRD12* showed significant associations with lower fluid intelligence scores ($\beta_{burden} = -0.028$, $P = 6.03 \times 10^{-10}$) and worse performance in the pairs matching task ($\beta_{burden} = 0.010$, $P = 2.93 \times 10^{-7}$). *GIGYF1* showed nominal associations with lower fractional anisotropy (FA) in the fornix tract ($\beta_{burden} = -0.059$, $P = 1.06 \times 10^{-4}$), and longer reaction time ($\beta_{burden} = 0.014$, $P = 1.40 \times 10^{-4}$). The Mendelian randomization analyses failed to uncover any causal relationship between cognition and alcohol consumption (Supplementary Data 25), in line with results from previous studies[29]. Given the limited evidence supporting causal links between cognition and alcohol consumption, it is plausible that the observed associations may stem from the pleiotropic effects of *ANKRD12* and *GIGYF1*.

The variant-phenotype association analyses revealed significant correlations with various white matter tracts. Notably, significant correlations were observed for FA in specific regions, including left

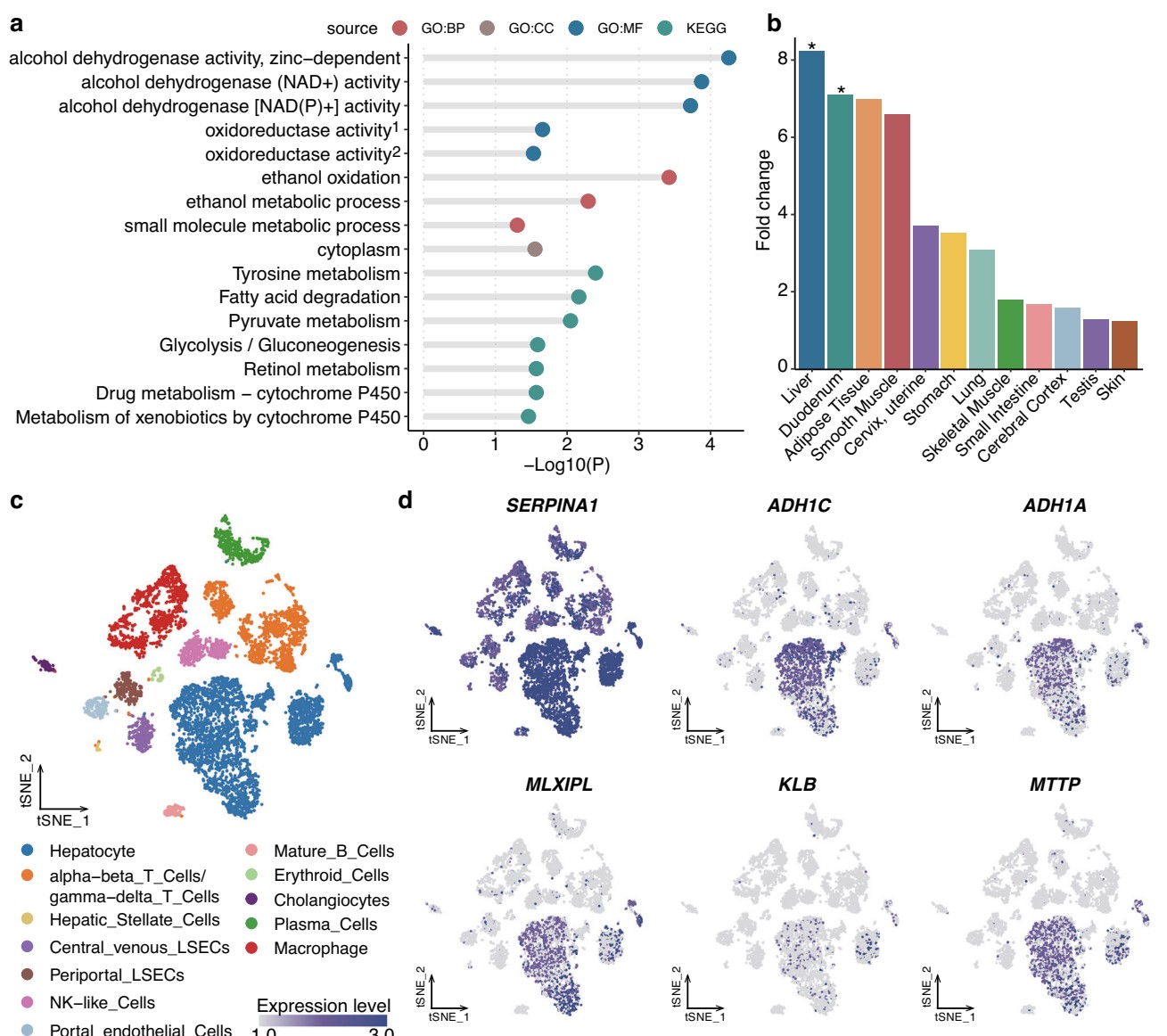

**Fig. 4 | Biological function of the alcohol consumption-related genes. a** Results of the functional enrichment analysis. The assessment of functional enrichment for the genes associated with alcohol consumption was assessed using the hypergeometric test. The g:SCS method was used for multiple testing correction. The $x$-axis indicates -$\log_{10}$ of the adjusted $p$-value for each term. The $y$-axis indicates different terms, and each source is marked with different colors. BP Biological process, GO Gene Ontology, MF Molecular function; KEGG, Kyoto Encyclopedia of Genes and Genomes. oxidoreductase activity[1], oxidoreductase activity, acting on the CH-OH group of donors, NAD or NADP as acceptor; oxidoreductase activity[2],

oxidoreductase activity, acting on CH-OH group of donors. **b** The bar plot shows the tissue-specific gene enrichment. The $x$-axis represents various tissue types. The $y$-axis indicates the fold-change values of the tissue-specific gene enrichment. Tissues with nominal enrichment were marked with *. **c** The t−Stochastic Neighbourhood Embedding (tSNE) plot shows different cell types within the liver. The cell type is represented by the color of the dots. **d** The feature plot displaying the expression level of tissue-specific genes in different cell types within the liver. *TPM* transcripts per million.

anterior limb of the internal capsule ($\beta = -0.072$, $P = 9.52 \times 10^{-13}$), genu of corpus callosum ($\beta = -0.069$, $P = 9.68 \times 10^{-13}$), and left superior frontal-occipital fasciculus ($\beta = -0.067$, $P = 6.13 \times 10^{-12}$).

## ExWAS in all white British participants and unrelated non-white British participants

Since SAIGE can handle sample relatedness in the regression model, we included all 373,152 white British participants (including both unrelated and related participants) in the analyses to increase statistical power. In the ExWAS for single variants, we identified 26 independent significant variants associated with alcohol consumption, including four variants not detected in the unrelated white British sample, of which two were not previously linked to

alcohol consumption (Supplementary Data 26). The gene-based collapsing analysis identified 23 potential alcohol consumption-related genes with an overall FDR < 0.05. Of the 23 genes, 13 were not found in the unrelated white British participants, and among these, eight were not previously associated with alcohol consumption (Supplementary Data 27).

Moreover, ExWAS was conducted in 61,076 unrelated non-white British participants. While the ExWAS for single variants identified one locus significantly linked to alcohol consumption (Supplementary Data 28), the gene-based collapsing analysis did not uncover any significant associations after FDR correction, potentially attributed to the constrained sample size among non-white British participants.

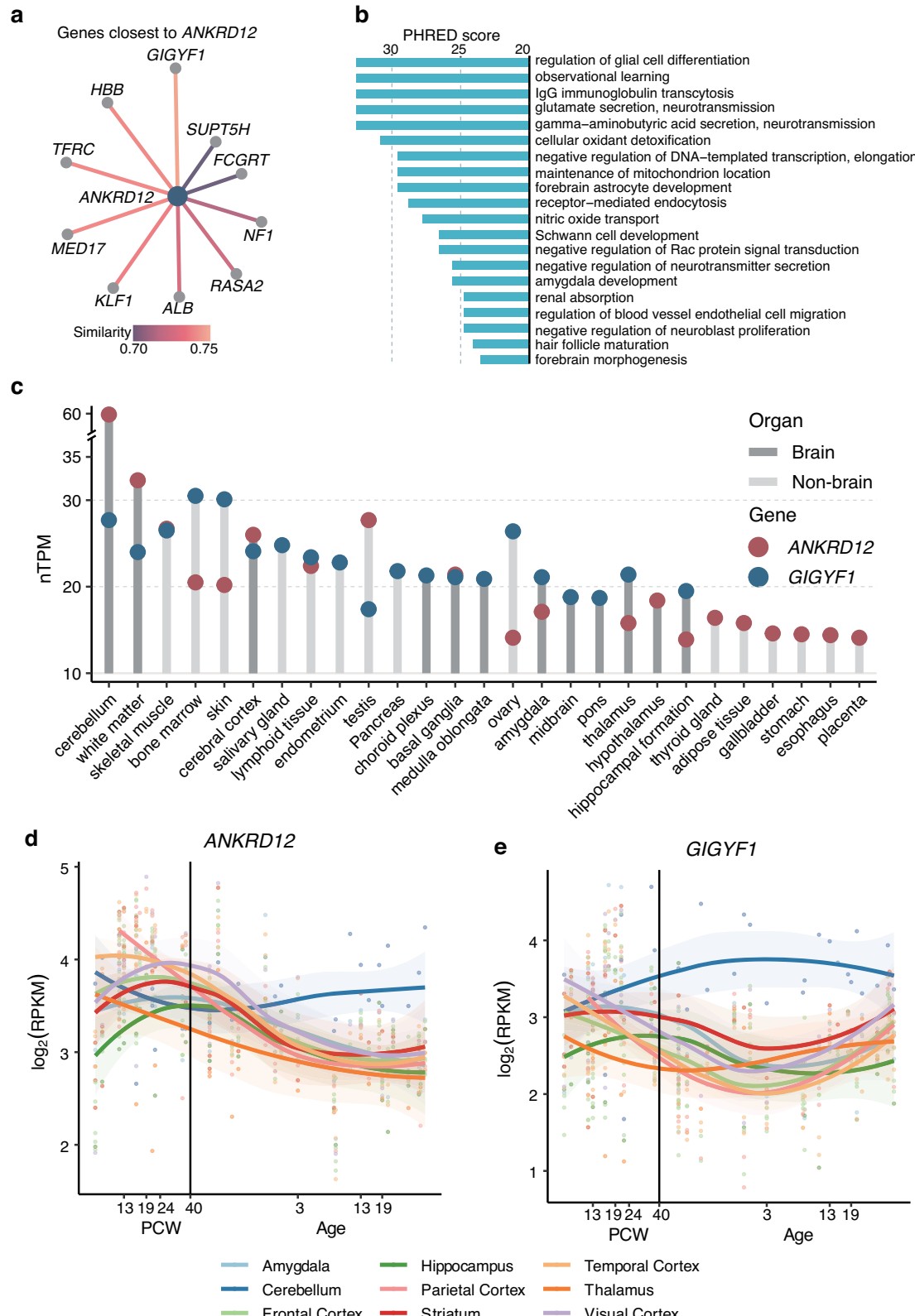

## Discussion

Herein we describe the largest comprehensive ExWAS of alcohol consumption to date and provide deep biological insights into the identified genes via functional analysis and phenome-wide association analysis with health-related data from the UKB. We identified ten previously unreported genes associated with alcohol consumption as well as replicated several known genes, which may shed light on pathophysiological processes in alcohol use. Furthermore, bioinformatics analyses supported the biological validity of the genetic associations and gene expression analysis highlighted the role of the cerebellum in alcohol consumption. PheWAS analyses provide strong support for the pleiotropic and consequent effects of alcohol consumption-related genes on human health, especially on inflammation, lipid metabolism, and white matter integrity.

**Fig. 5 | Functional analysis of the rare-variant genes identified in our study.**
**a** Top 10 genes with the most similar quantitative trait associations to *ANKRD12* using exome-sequencing data in the UKB. **b** Enriched gene sets of *ANKRD12* plus the top 10 similar genes of ANKRD12. Fisher's exact test was performed by Gene-SCOUT. The *x*-axis indicates the PHRED score (−10×log₁₀(P)), derived from Gene-SCOUT. **c** Expression of *ANKRD12* and *GIGYF1* in human tissues from the Human Protein Atlas. Abbreviations: nTPM, normalised transcripts per million. The top 20 tissues are included here; the complete plot is available in Supplementary Fig. 23. **d** Lifespan spatiotemporal expression trajectory of *ANKRD12* in the human brain. Expression is shown in both prenatal and postnatal periods derived from mRNA-seq data of the PsychENCODE study [25]. The *x*-axis denotes the age, represented in both post-conception weeks (prenatal) and years (postnatal), categorized into eight

distinct periods: < 13 post-conception weeks (PCW)), 13-18 PCW, 19-23 PCW, 24-37 PCW, 0-2 years, 3–12 years, 13–19 years, and >19 years. The *y*-axis depicts the log₂-transformed expression value, given in reads per kilobase million (RPKM). Each brain region's expression trajectory was visualized through a fitted non-linear LOESS regression line, accompanied by error bands (shaded areas) indicating the 95% confidence interval. **e.** Lifespan spatiotemporal expression trajectory of *GIGYF1* in the human brain. The *x*-axis denotes the age, represented in both post-conception weeks (prenatal) and years (postnatal). The log₂ transformed expression values were represented on the *y*-axis. Each brain region's expression trajectory was illustrated through a fitted non-linear LOESS regression line, accompanied by error bands (shaded areas) denoting the 95% confidence interval.

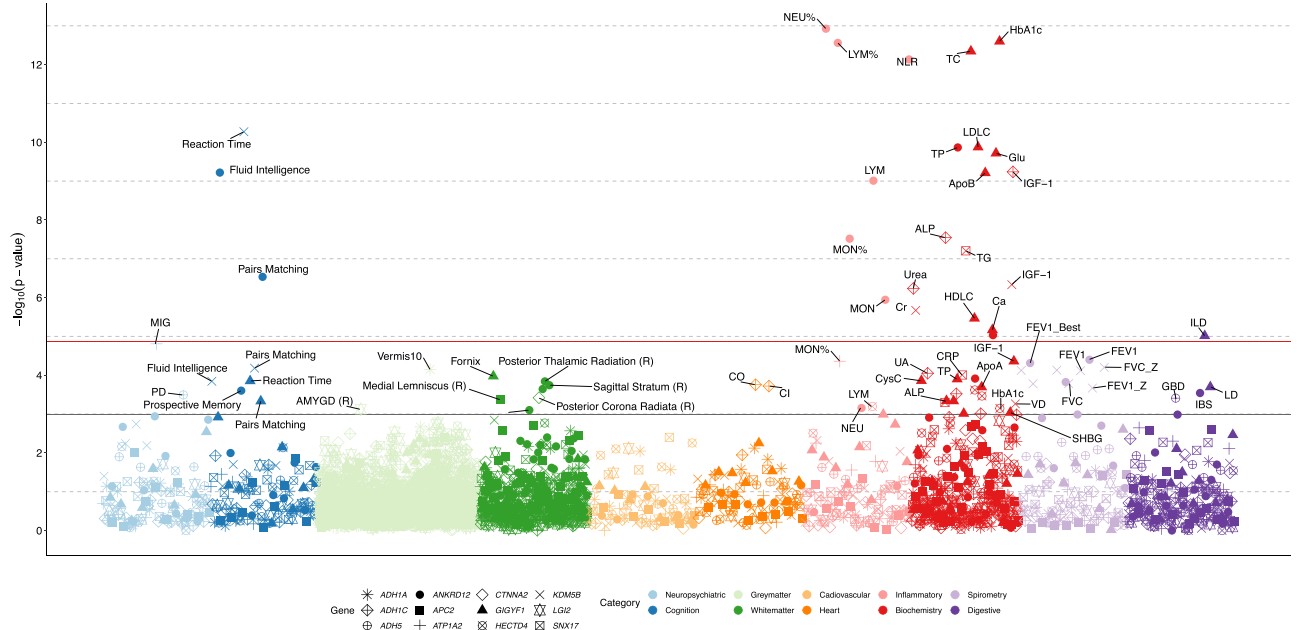

**Fig. 6 | Phenotypic associations of the rare-variant genes linked to alcohol consumption.** The *x*-axis represented various categories encompassing various phenotypes, including 10 neuropsychiatric diseases, 7 cardiovascular diseases, 19 digestive diseases, 10 cognition scores, 9 inflammatory traits, 30 blood biochemistry traits, 214 nervous traits (166 grey matter measures and 48 white matter measures), 8 heart structure measures, and 9 spirometry measures. For each phenotype-gene association, we applied three different maximum MAF cutoffs (0.01%, 0.1% and 1%) and two variant annotations (LOF and LOF+missense). The *y*-axis indicates the -log₁₀ of the p-value for each association, with p-values adjusted for age, sex, and ten ancestral principal components. The red horizontal line denotes the threshold for significant association ($P < 0.05/316/12 = 1.32 \times 10^{-5}$), and the grey line signifies the threshold for a significant association to a lesser extent ($P < 0.001$). Presented p-values are two-sided and unadjusted for multiple testing. For each phenotype-gene association, the minimum p-value was plotted. Results for all genes at different variant frequencies and groups are presented in Supplementary Data 23. NEU% Neutrophill Percentage, HbA1c Glycated Haemoglobin (Hba1C), LYM% Lymphocyte Percentage, TC Cholesterol, *NLR* Neutrophill Lymphocyte Ratio, Reaction Time Mean Time To Correctly Identify Matches, LDLC Ldl

Cholesterol, TP Total Protein, Glu Glucose; IGF-1, IGF1; Fluid Intelligence, Fluid Intelligence Score, ApoB Apolipoprotein B, LYM Lymphocyte Count, ALP Alkaline Phosphatase, MON%, Monocyte Percentage; TG, Triglycerides; Pairs Matching, Number Of Incorrect Matches In Round; Urea, Urea; MON, Monocyte Count; Cr, Creatinine, HDLC Hdl Cholesterol, Ca Calcium, ILD Inflammatory Liver Disease, MIG Migraine, FEV1 Forced Expiratory Volume In 1-Second (Fev1), FEV1_Best, Forced Expiratory Volume In 1-Second (Fev1), Best Measure; FVC_Z, Forced Vital Capacity (Fvc) Z-Score, Vermis10 Vermis_10, FEV1_predperc Forced Expiratory Volume In 1-Second (Fev1), Predicted Percentage; UA, Urate; CRP, C-Reactive Protein; Fornix, Fornix; CysC, Cystatin C; Posterior Thalamic Radiation (R), Posterior Thalamic Radiation (R); FVC, Forced Vital Capacity (Fvc); FVC_Best, Forced Vital Capacity (Fvc), Best Measure; CO, Cardiac Output, Sagittal Stratum (R), Sagittal Stratum (R); CI, Cardiac Index; LD, Liver Disease; ApoA, Apolipoprotein A, FEV1_Z Forced Expiratory Volume In 1-Second (Fev1) Z-Score, Posterior Corona Radiata (L), Posterior Corona Radiata (L), IBS Inflammatory Bowel Disease; PD, Parkinson's Disease; GBD, Gallbladder Disease; Alb, Albumin, VD Vitamin D, NEU Neutrophill Count, AMYGD (R), Volume of right amygdala.

Previous GWAS studies have enabled the identification of alcohol consumption-related genes, but our study extended previous findings via the discovery of more genes as well as the identification of more common and rare variants to the reported genes of alcohol consumption. We have identified thirteen genes at exome-wide significance based from rare variants using gene-based collapsing analysis, seven of which (*GIGYF1*, *ANKRD12*, *KDM5B*, *APC2*, *LGI2*, *ENSG00000224076* and *ATP1A2*) were not reported by previous GWAS studies. Moreover, among the 174 reported genes from the most

recent GWAS studies[6,7,16,17], twenty-five showed nominal significance and *ADH1C* passed Bonferroni correction. Notably, utilizing the LOVO analysis, we found that for those reported GWAS genes including *ADH1C*, *ADH1A*, *SNX17*, and *ADH5*, removal of a single SNP leads to loss of significance in gene-based collapsing analysis, while for the genes not previously reported in the GWAS studies, removal of any single SNP does not influence the significance. The results indicated that the significance of these genes is the cumulative effect from a group of rare SNPs, which may explain why they were not detected by previous

GWAS studies. This is further validated by the single variant analysis, where a significant signal was detected in *ADH1C* while not in those genes. Our results emphasized the value of rare variants as well as the necessity of gene-based collapsing analysis in WES studies on alcohol consumption.

For the two rare-variant genes (*GIGYF1* and *ANKRD12*) associated with alcohol consumption, *GIGYF1*, identified as a risk gene for diabetes in earlier research[13,30], is a protein-coding gene intricately involved in the regulation of cell growth and division. One meta-analysis of 38 studies demonstrated that a moderate level of alcohol intake was linked to a lower risk of type 2 diabetes compared to abstainers[31]. The association might be mediated by the beneficial metabolic effect of alcohol consumption such as altered HDL cholesterol and inflammation levels[26]. Meanwhile, results of the PheWAS showed that the alcohol consumption-related gene, *GIGYF1*, was significantly associated with blood levels of HDL cholesterol and several inflammatory biomarkers. Therefore, *GIGYF1* may participate in the metabolic disturbance caused by alcohol consumption. As for another gene *ANKRD12*, less evidence was found on its possible role in alcohol consumption. While the Gene-SCOUT analysis provided interesting findings that *GIGYF1* and *ANKRD12* showed high similarity in biomarker profiles, which suggested that they might execute similar biological functions. Interestingly, *ANKRD12* and *GIGYF1* are associated with a higher Townsend deprivation index, which could possibly lead to a less access to alcohol[32]. Given that those genes were associated with cognitive function in our PheWAS results and in previous studies[33,34], it is possible that reduced cognitive function in the gene carriers results in increased material deprivation and in turn reduced alcohol consumption. Nevertheless, the findings may be confounded by many factors and the causality is not validated by mechanism study, so further research is needed to clarify the potential associations of *GIGYF1* and *ANKRD12* with alcohol consumption.

In addition to the discovery of genetic associations, we also provide insights into alcohol metabolism-related brain alterations based on the two rare-variant genes identified in this study. The Gene-SCOUT analysis identified a series of genes that were highly similar to these two genes. These genes displayed significant enrichment in the regulation of glial cell differentiation and observational learning. As evidenced by previous human and animal studies, disrupted differentiation of glial cell (astrocytes and oligodendrocytes) is one of the human alcohol-related neuropathology[35] and heavy alcohol exposures could result in cognitive impairment[36]. Since alcohol consumption influences intracellular signaling mechanisms, causing alterations in gene expression that gradually produce long-lasting damage in the brain[37], these identified genes might be involved in the pathological process. What's more, glia dysfunction is known to cause white matter atrophy, and these two genes are significantly expressed in white matter, further hinting that they might mediate alcohol-related brain damage. Another finding lies in their dominant expression in the cerebellum, one of the major target organs of alcohol abuse. Moreover, *ANKRD12* and *GIGYF1* are well-known genes for reduced cognitive function and intellectual disability as evidenced by previous studies[33,34]. Consistently with previous findings, our PheWAS analyses indicated strong correlations between these genes and cognitive decline as well as altered white matter integrity, which suggests that these genes play a significant role in brain function and structure. The findings are plausible as prior studies have observed the associations between heavy alcohol consumption and changes in brain structure[38,39]. More interestingly, alcohol consumption-related white matter microstructure changes have been considered a hallmark of AUD[40,41]. Therefore, *ANKRD12*, significantly associated with alcohol consumption, AUDIT, and white matter integrity alterations, might serve as therapeutic targets for the prevention of AUD.

We observed sex heterogeneity for *KDM5B*, the association between *KDM5B* and alcohol consumption was only observed in the male group. As we only observed heterogeneity in one gene, it is possible due to the sex-specific biological function of *KDM5B*. *KDM5B* encodes a lysine-specific histone demethylase, which is an important regulator of liver molecular pathways after alcohol consumption[42]. Previous studies found sex-specific roles of KDM5B in the alcohol-induced hepatic response, which regulates a fibrogenic program in females while contributes to hepatocyte dedifferentiation and fatty acid synthesis in males[43,44]. However, the sex-specific mechanisms underlying the influence of *KDM5B* on alcohol consumption is still unclear. Future studies to identify the mechanisms will be necessary.

Despite these significant findings, our study has some limitations. First, as WES could only detect variants in the protein-coding regions, the possible genetic associations in non-protein-coding region were less investigated in this work. Second, because of the scarcity of a comparable population cohort with genetic sequencing and phenotype data for replication, we relied on existing GWAS data for alcohol consumption and AUD to support our findings. Further whole-exome studies are needed to replicate the identified genes. Third, the causality between the reported genes and alcohol use was largely unknown. Further research are needed to replicate and verify the identified genes and the potential relationship with alcohol consumption. Lastly, participants who drinking up to 3 times monthly and less were assigned a weekly drinking level of zero following a previous study[45]. While this simplified approach may introduce some error into their drinking levels, it is expected to be relatively small given the infrequency of their alcohol consumption.

In conclusion, by sequencing the protein-coding regions, we were able to replicate the genes previously reported and identify common and rare coding variants that have a strong effect on alcohol consumption. Additionally, functional analysis of the identified genes not only recapitulated known biological processes in alcohol consumption but also provided insights into the brain's role in alcohol consumption. We anticipate that our findings of the alcohol consumption-related genes will facilitate the identification of individuals that are vulnerable or intolerant to alcohol consumption, contributing eventually to the prevention as well as treatment of alcohol-related adverse outcomes.

## Methods

### UK Biobank
The UKB included phenotypic and genetic information for approximately 500,000 participants of ages between 40 and 69[46,47]. Informed consent has been signed by all participants. The UKB cohort was approved by the NHS National Research Ethics Service North West (reference number: 16/NW/0274). The data utilized in the study included demographic data, alcohol-related phenotypes, neuropsychiatric diseases, cardiovascular diseases, cognition, brain grey matter and white matter phenotypes, heart function, lung function, biochemistry, and inflammation phenotypes. The research was performed under application number 19542.

### Study phenotypes
The alcohol consumption score was determined through a self-administered touchscreen interview conducted during the baseline appointment. Initial data acquisition involved obtaining mean weekly alcohol consumption data, taking into account various beverage types, from participants reporting alcohol consumption more than once or twice weekly. Each alcoholic drink type was measured in specific units: spirits in measures, wines in glasses, and beer/cider in pints, approximately equating to one, two, and two point five units, respectively. For respondents indicating intake frequencies of "one to three times a month," "special occasions only," or "never" (for whom weekly alcohol consumption data were unavailable), a weekly volume of 0 units was assigned. The determination of alcoholic units per week involved aggregating the intakes for these five drink types, consistent with a previous study[45]. The median alcoholic units per week of the whole

sample was 10 (Supplementary Data 2). The alcohol consumption score was the log (units+1) transformed alcoholic units per week. Detailed information was available in Supplementary Data 1.

## Whole exome sequencing data

WES was performed for approximately 454,756 individuals from the UKB with IDT xGen Exome Research Panel v1.0[11,18]. We implemented centralized quality control following extensive quality control procedures following previous research[13]. Concisely, multi-allelic sites were segregated into bi-allelic sites and calls with poor genotype quality or excessively low/high genotype depth were marked as no-call. Next, we excluded variants located in Ensembl low-complexity regions, along with variants possessing call rate $\leq 90\%$, and Hardy-Weinberg Equilibrium (HWE) $P$-value $\leq 10^{-15}$. Finally, we removed participants who withdrew from the UKB, duplicates, participants exhibiting discrepancies between self-reported and genetically indicated sex, and participants with Ti/Tv, Het/Hom, SNV/indel, and the amount of singletons exceeding 8 standard deviations from the mean. Additionally, we excluded individuals who were genetically related at the 3rd degree or closer in the main analysis. Overall, a total of 304,119 individuals with available alcohol consumption data and genetic data passed the initial quality check and were used in the main analysis. We additionally conducted ExWAS in all (both genetically related and unrelated) white British participants and unrelated non-white British individuals. White British individuals were identified as the intersection of participants who self-reported as 'White British' and those who exhibited very similar genetic ancestry based on genetic components. To control population stratification, we generated the top 10 ancestral principal components (PCs) using a high-quality independent autosomal variants subset, as outlined in a prior study[13]. Specifically, this subset of variants comprised variants with MAF > 0.1%, HWE $P > 10^{-6}$, missingness < 1%, and underwent two rounds of pruning (--indep-pairwise 200 100 0.1 and 200 100 0.05 in PLINK).

## Variant annotation

First, rare variants were defined as MAF less than 1%. SnpEff was utilized to annotate the variants[48], during which the most detrimental consequence of the gene transcript was retained. Subsequently, variants annotated as frameshift, splicing donor, stop gain, splicing acceptor, stop loss, and start loss were categorized as loss of function (LOF). Variants that were consistently predicted as deleteriousness in SIFT[49], PolyPhen2 HDIV, and PolyPhen2 HVAR[50], LRT[51], and MutationTaster[52] were defined as likely deleterious missense.

## ExWAS

ExWAS analysis was conducted using the SKAT-O test through SAIGE-GENE + [53]. In SAIGE-GENE +, ultra-rare variants (minor allele carrier (MAC) $\leq 10$) were collapsed into a pseudo marker, effectively addressing data sparsity caused by the presence of ultra-rare variants[53]. Therefore, both rare and ultra-rare variants could be investigated. First, single-variant association analyses were performed for all variants with MAC $\geq 20$, as suggested by SAIGE-GENE + [53]. Independent significant variants were identified using linkage disequilibrium (LD)-clumping ($r^2 < 0.1$), with the UKB WES data utilized as the reference panel, and subsequently mapped to genes using VEP[54]. Then, in the gene-based collapsing analyses, SKAT-O tests were conducted utilizing the minimum p-value method[53,55]. We used three distinct maximum MAF cutoffs (0.01%, 0.1%, and 1%) and two annotations masks (LOF and LOF plus missense). We adjusted age, sex, and the top ten ancestral PCs (which were calculated with WES data). All quantitative phenotypes underwent inverse normalization in SAIGE-GENE +. A relative coefficient cutoff of 0.05 was applied to the sparse genetic relationship matrix for the estimation of variance ratios.

## Genotype and imputation

Genotype data (version 3) were from the UKB cohort. The UKB conducted array design, genotyping, quality control, and imputation procedures[46]. We performed quality control (excluding variants with MAF < 0.005, INFO < 0.3, call rate < 90% or HWE $P < 10^{-50}$) with PLINK v2[56] software. Additionally, participants with missingness less than 0.05, no sex mismatch, no abnormal sex chromosome aneuploidy, no outliers in heterozygosity rate, and estimated white British ancestry, with a maximum of ten putative third-generation relatives, were incorporated into the analysis.

## ExWAS for AUDIT

To extend the implications of alcohol consumption findings to alcohol use disorder, we conducted an ExWAS utilizing measures from the Alcohol Use Disorders Identification Test (AUDIT)[57], obtained through an online mental health questionnaire and processed following the methodology detailed in the previous study[58]. Specifically, the scores for the AUDIT subdomains, representing alcohol consumption (AUDIT-C) and indicating alcohol dependence and problematic alcohol use (AUDIT-P), were calculated by consolidating scores from items 1–3 and items 4–10, respectively. The total score (AUDIT-T) was the sum of items 1–10. Detailed information was available in Supplementary Data 1. A total of 101,240 participants with available AUDIT measurements, WES data and covariate information were used for the analyses. We conducted ExWAS for both the total score and the subscores.

## LOVO analysis

The LOVO analysis was performed for associations identified in the gene-based analysis. For each gene-phenotype association, the collapsing test was iterated upon excluding each variant initially included, where each variant would have a P-value. This was undertaken to address specific aspects: firstly, to examine the stability and consistency of the results across variant exclusions; secondly, to discern whether the gene-based collapsing association results were predominantly driven by specific variants; and finally, to investigate whether the observed gene-based collapsing associations were influenced by numerous rare variants characterized by relatively small effect sizes. If the collapsing analysis after removing a single variant yields an attenuated significance ($P > 0.01$), that single variant was considered to predominantly drive the gene-phenotype association[13]. This analytical approach allows for a comprehensive evaluation of the role of individual variants within the broader gene-based context.

## Conditional analysis

To test for independence between the significant rare variant associations and nearby common variation, we re-conducted the gene-based collapsing analyses additionally correcting the nearby common variants associated with alcohol consumption[13]. First, we conducted association analyses for common variants (MAF > 0.5%) within the 500 kb genomic region of the identified genes, utilizing the UKB imputed genotype data. Then, LD-clumping was performed to identify independent significant loci ($P < 1 \times 10^{-5}$ and $r^2 < 0.01$). At last, we performed the collapsing analyses additionally adjusting for the independent significant loci.

## Burden heritability estimation

We estimated the burden heritability based on rare coding variants (LOF and missense) using the burden heritability regression (BHR) method[59]. The BHR performed regression of the burden test statistic on the burden score using summary statistics of the association analysis and allele frequencies at the variant level, and derived the burden heritability through estimation of the regression slope[59].

## Pathway enrichment analysis

We used the g:Profiler[60] software to conduct the enrichment analysis, selecting Gene Ontology and KEGG database as the gene set databases. The g:SCS (Set Counts and Sizes) correction method was employed for multiple testing correction.

## Tissue enrichment and expression analysis

To determine whether the identified genes were enriched in multiple tissues, we conducted tissue enrichment analysis using the R package TissueEnrich[22]. The source data were from the Human Protein Atlas, and the hypergeometric test was used[22].

Transcript expression levels of the two genes (*GIGYF1* and *ANKRD12*) in 256 tissues were determined utilizing RNA sequencing data from the Human Protein Atlas[24]. The dataset corresponds to Human Protein Atlas version 22.0 and Ensembl version 103.38. Additional details regarding the data are available elsewhere at (https://www.ebi.ac.uk/biostudies/arrayexpress/studies/E-MTAB-2836).

## Lifespan spatio-temporal gene expression trajectory

The lifespan spatio-temporal brain expression trajectories of the alcohol consumption-related genes were characterized using the mRNA-seq data of human brain from the PsychENCODE study[25]. The expression of each gene in each anatomical tissue was estimated. Gene expression levels was quantified utilizing the reads per kilobase per million mapped reads (RPKM) metric.

## Single-cell expression

We used liver scRNA-seq data from Gene Expression Omnibus (GEO) database (accession ID: GSE115469)[61] and processed it with the R package Seurat[62]. Individual cells with low quality, defined as the cells with less than 200 expressed genes or larger than 75% mitochondrial counts, were excluded. Then the gene expression matrix underwent normalization using the NormalizeData function in Seurat[62]. The top 25 PCs and a resolution of 0.4 were used to conduct clustering, and then the clusters were annotated according to the previous publication[61].

Additionally, the brain scRNA-seq data sourced from temporal cortex tissues was obtained from the GEO database under accession ID GSE173731[63]. In the dataset, all cell types in the brain were isolated and sequenced[63]. Analysis and visualization were performed using the metadata files with the R package Seurat[62].

## Gene similarity

We utilized Gene-SCOUT[23] to estimate the similarities between genes using association results of collapsing analyses across various quantitative traits in the UKB. In this tool, we searched the "seed gene" *ANKRD12* to identify the similar genes. The top 10 similar genes and the "seed gene" were then employed in the enrichment analysis with Gene Ontology terms[23].

## MRI data and preprocessing

Structural MRI data were obtained from three dedicated and identical imaging centers[64,65]. Preprocessing of this data followed a pipeline established in previous studies[66,67] with SPM12 software and the CAT12 toolbox[68] with default settings. This included high-dimensional spatial normalization, nonlinear modulations, and smoothing (with an 8 mm half-maximum full-width Gaussian kernel). For regional grey matter volume, we employed the Automated Anatomical Labeling 3 (AAL3) atlas[69], a brain parcellation system that subdivides the brain into 166 distinct regions. We utilized the AAL3 atlas due to its finer parcellation, especially in the subcortical regions, which are closely linked to alcohol use and addiction.

We utilized fractional anisotropy (FA) of white matter tracts provided by UKB. Detailed data processing and quality control procedures have been comprehensively outlined in prior study[60]. Specifically, dual diffusion-weighted shells were employed to acquire diffusion-weighted images, incorporating 50 distinct diffusion-encoding directions for each shell, and with a resolution of $2 \times 2 \times 2$ mm. TBSS[70] was used to conduct the alignment of FA images to a standard-space white matter skeleton. FA images was further improved with high-dimensional FNIRT-based warping for enhanced alignment[71]. Our analyses encompassed 48 distinct white matter tracts extracted based on the JHU ICBM-DTI-81 atlas[72].

## Phenome-wide association analysis

The phenotypes in PheWAS were centered around traits that are associated with alcohol consumption, including behavioral aspects and health outcomes. The disease-related analysis covered neuropsychiatric diseases, cardiovascular diseases, and digestive diseases, which can be impacted by alcohol consumption patterns. Additionally, the analysis incorporated cognitive tasks, inflammatory traits, blood biochemistry traits, neuroimaging traits (including grey and white matter measures), and cardiac and lung function measures, all of which are pertinent to understanding the impacts of genes related to alcohol consumption on human health and functioning. This comprehensive selection of phenotypes aligns with the aim of investigating the potential genetic influences on alcohol consumption and its related health implications. In the analysis of diseases, we investigated 10 neuropsychiatric diseases, 7 cardiovascular diseases, and 19 digestive diseases. For the analysis of continuous phenotypes, we examined 10 cognition tasks, 9 inflammatory traits, 30 blood biochemistry traits, 214 neuroimaging traits (including 166 grey matter measures and 48 white matter measures), 8 heart structure measures, and 9 spirometry measures. Comprehensive details regarding the phenotypes can be found in Supplementary Data 22. We used single-variant association tests for identified variants and SKAT-O tests for identified genes[53], adjusting for the top ten ancestral PCs, age, and sex.

For the cognitive function tasks, data were preprocessed similar to the previous study[73]. We incorporated cognitive tests from both baseline and imaging follow-up. Specifically, we selected the timepoints that corresponded to the maximum sample size for each cognitive test.

## Mendelian randomization analysis

To explore the mediating relationships between *ANKRD12*, *GIGYF1*, cognition, and alcohol consumption, we first conducted a bidirectional Mendelian randomization (MR) between cognition and alcohol consumption using TwoSampleMR R package. We employed GWAS summary data for the general factor of intelligence, derived from a compilation of seven distinct cognitive tests[74], all sourced from the UK Biobank. Ensuring the avoidance of sample overlap, we utilized separate GWAS summary data for alcohol consumption, excluding participants from the UK Biobank[19].

## Sensitivity analysis

To evaluate the stability of the main results, we conducted multiple sensitivity analyses. Initially, we excluded participants who were former drinkers and non-drinkers (Field 20117) and performed association analysis for the identified genes. Additionally, we adjusted for rs1229984, a well-known alcohol consumption-related locus[6,21], to identify independent associations.

## Reporting summary

Further information on research design is available in the Nature Portfolio Reporting Summary linked to this article.

# Data availability

The data used in the study from the UKB were accessible under restricted access (application number 19542). Access can be procured by submitting an application through the UKB platform (https://www.ukbiobank.ac.uk/). The scRNA-seq data are documented in the GEO database, accessible under accession codes GSE115469 and GSE173731.

The transcript expression data are accessible in the Human Protein Atlas database (https://v22.proteinatlas.org/about/download). The processed human brain mRNA sequencing data were available in the PsychENCODE study (http://development.psychencode.org/files/processed_data/RNA-seq/). Summary GWAS statistics from FinnGen are available at https://storage.googleapis.com/finngen-public-data-r9/summary_stats/finngen_R9_AUD.gz and https://storage.googleapis.com/finngen-public-data-r9/summary_stats/finngen_R9_AUD_SWEDISH.gz. KEGG database used in gProfiler are available at https://www.genome.jp/kegg/pathway.html. The Gene Ontology database used in gProfiler are available at https://geneontology.org/docs/download-ontology/. The paper and/or the Supplementary Information contain all necessary data to assess the conclusions. In addition, this paper includes source data. Source data are provided with this paper.

## Code availability

ExWAS analyses and PheWAS analyses was performed via the R package SAIGE GENE+ which was available on https://github.com/saigegit/SAIGE. Burden heritability regression analysis was performed via the R package BHR (v.0.1.0, https://github.com/ajaynadig/bhr). Annotation of significant variants was conducted with SnpEff (https://pcingola.github.io/SnpEff/). Gene ontology enrichment analysis was conducted using g:Profiler (https://biit.cs.ut.ee/gprofiler/gost) and tissue enrichment analysis was performed via the R package TissueEnrich (v.1.16.0, https://github.com/Tuteja-Lab/TissueEnrich). The scRNA-seq data were analyzed and visualized using the R package Seurat using the R package Seurat (v.4.3.0, https://satijalab.org/seurat/index.html). Gene-Scout was available through the website: https://astrazeneca-cgr-publications.github.io/gene-scout/.

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

## Acknowledgements

We express our gratitude to the participants of the UK Biobank for their valuable time, and we extend our appreciation to the dedicated team members of the UK Biobank for their efforts in data collection. We acknowledge the participants and investigators involved in the FinnGen study. We also acknowledge the contributions of MacParland, S.A. et al. and Garcia, F.J. et al. for providing the scRNA-seq matrix. We acknowledge the Human Protein Atlas project, the PsychENCODE Consortium and the FinnGen project for their unwavering dedication to advancing scientific research. W Cheng received support through grants from the

National Natural Sciences Foundation of China (no. 82071997) and the Shanghai Rising-Star Program (no. 21QA1408700). J.T. Yu received support through grants from the Science and Technology Innovation 2030 Major Projects (2022ZD0211600), National Natural Science Foundation of China (82071201, 81971032, 92249305), Shanghai Municipal Science and Technology Major Project (No.2018SHZDZX01), Research Start-up Fund of Huashan Hospital (2022QD002), Excellence 2025 Talent Cultivation Program at Fudan University (3030277001), Shanghai Talent Development Funding for The Project (2019074), and ZHANG-JIANG LAB, Tianqiao and Chrissy Chen Institute, and the State Key Laboratory of Neurobiology and Frontiers Center for Brain Science of Ministry of Education, Fudan University. J.F. Feng received support through grants from National Key R&D Program of China (No. 2018YFC1312904 and No. 2019YFA0709502), the Shanghai Municipal Science and Technology Major Project (No. 2018SHZDZX01), the 111 Project (No. B18015), Shanghai Center for Brain Science and Brain-Inspired Technology and Zhangjiang Lab. T.Y. Jia received support through grants from the National Key R&D Program of China (No. 2019YFA0709501) and the National Natural Science Foundation of China (T2122005, No. 81801773).

## Author contributions

All authors had complete access to the data in this study and acknowledged the responsibility for its submission for publication. W.C., J.T.Y., and J.F.F. designed the study. J.J.K. and Y.T.D. conducted the main analyses and drafted the manuscript. B.S.W., Z.Y.L., W.S.L., S.T.X., L.Y., and J.Y. contributed to data collection and analyses. X.H.G. and T.Y.J. contributed to data interpretation. J.T.Y., W.C., and J.F.F. provided critical revisions to the manuscript. All authors have reviewed and approved the final version.

## Competing interests

The authors declare no competing interests.
