## [Peer Review File · Nature Communications]

Whole exome sequencing analysis identifies genes for alcohol consumptionReviewers' Comments:

Reviewer #1:

Remarks to the Author:

This manuscript presents the results of an exome-wide association study (ExWAS) conducted on 365,196 individuals from the UK Biobank, aimed at uncovering novel genes associated with alcohol drinking per week (DPW). This study identified 34 genes, of which 16 are novel. Among these, 14 were also associated with alcohol use disorder in FinnGen. Notably, rare predicted loss-of-function variants in two novel genes, GIGYF1 and ANKRD12, collectively indicated detrimental effects on DPW. Overall, this is an engaging paper with significant results.

However, I have noted the following points for improvement:

The manuscript has multiple typographical and grammatical errors. Please correct "Notably," in line 46 and eliminate the redundant sentence in lines 230 and 231. Additionally, review the sentences around lines 286-287 and 310-311 for clarity. I couldn't pinpoint all sentences with potential issues, but a thorough review is necessary.

In line 126, it would be beneficial to provide more context or explanation for the phrase "As expected."

On line 140, the phrase ", which was the highest among the different groups," is unclear. Please provide more context or clarification.

In line 184, instead of referring to them as "the two novel genes," please specify the names of the genes for clarity.

The first sentence of the paragraph at line 225 is unclear. Please revise for better understanding.

On line 264, I suggest adding "but" before "our study" to improve the flow and coherence of the sentence.

In Figure 2 (A) and (B), the differentiation between circles and squares is not clear. Please consider improving this for better comprehension.

Reviewer #2:

Remarks to the Author:

In this study, the authors undertake a GWAS and ExWAS analysis of alcohol use using several metrics. Their approach is sound and the methods are valid.

This is the first ExWAS report of for this trait in UKB that I am aware of, so it has some novelty. Of the highlighted "novel" genes, ANKRD12 and GIGYF1 are well known genes for reduced cognitive function. GIGYF1 has a multitude of additional associations, as mentioned in this manuscript. Please see references below suggesting these are primarily cognitive dysfunction / intellectual disability genes.

My major concern is the idea that these are genes primarily related to alcohol consumption in some way, or that some of the PheWAS associations may be mediated by alcohol use disorder. I would recommend the authors leverage MR to understand whether cognitive function / income / Townsend deprivation index (all available in UKB) alterations mediate reduced alcohol use (likely) or vice versa (would be surprising). This could be done with both the rare variants of interest (in GIGYF1 and ANKRD12) and common variants more broadly related to these traits.

Otherwise, I think the study is reasonably well executed, and other than some minor grammatical errors or typos, reasonably well written.

Refs:

ANKRD12 & cognition:

<https://www.nature.com/articles/s41588-023-01398-8>

GIGYF1 and cognition:

<https://www.nature.com/articles/s41588-023-01398-8>

<https://www.nature.com/articles/s41588-022-01104-0/figures/3>

<https://www.deciphergenomics.org/gene/GIGYF1/overview/clinical-info>

Reviewer #3:

Remarks to the Author:

This study performed whole exome sequencing analysis of alcohol consumption in UKB. As the authors point out, most previous studies have focused on common variants. Overall this study is of high interest and relevance, and offers novel insights into the mechanisms by which alcohol impacts the body. The authors found 16 novel genes associated with alcohol, including 2 novel LOF genes. They also performed several additional analyses, including cell enrichment, PhWAS which adds to the story. Particularly interesting are associations with genes linked to alcohol metabolism (ADH1C), enrichment in brain regions impacted by alcohol e.g. cerebellum, and sex differences.

Specific comments below. As I do not have specific expertise on rare variant analyses, I have focused the critique to areas within my knowledge:

1. Abstract -

i) "disturbance of white matter integrity" - not sure this can be inferred given the measures are cross-sectional.

ii) ".showed deleterious effects on DPW" - I was not clear which direction this meant.

2. Methods-

i) How were European ancestry participants identified? It appears to be self-reported ("estimated ancestry") - surely using genetic ancestry would be more robust?

ii) I think giving a rationale for use of different alcohol phenotypes e.g. AUDIT measures is important.

iii) DPW in alcohol papers usually means drinks per week (US drinks which do not equate to UK units). Here it appears to mean log-transformed UK units? Terminology is confusing.

iv) Drinking up to 3 times monthly was assigned weekly volume of 0 - this does not seem correct?

v) What was done with former drinkers? Many are "sick quitters" so can influence results and need to be handled with care in analyses.

vi) Were non-drinkers excluded? I was not clear.

vii) For the PhWAS it was not clear to me how/why the phenotypes were selected? This includes the MRI phenotypes. What was corrected for in this analyses? "214 nervous traits" I think refers to MRI phenotypes? If so then a there is probably a better term.

3. Results -

i) Average DPW = 1.98? Again, see point above, this is confusing as I originally read as drinks per week and thought it was wrong as too low. I think giving drinks/units per week is more informative for reader.

ii) What "white matter measures" in the corpus callosum were examined/associated?

4. Discussion-

i) Grammatical errors in a couple of places, which looked out of place in an otherwise nicely written

paper.

5. Supplement -

- i) It seems that the authors ran their own pre-processing MRI pipeline - what was the rationale for this when UKB imaging has been pre-processed and QC'd by team and IDPs used by most other publications? There is no detail about the white matter measures anywhere I could see in the paper.
- ii) Cognitive tests - which timepoints were used for tests? Seems like a mixture.
- iii) SFigures: SFig 10-14 the text is unreadably small, SFig 19 does not spell out abbreviations, SFig 25 does not spell out abbreviations and some text unreadable as overlapping.

Reviewer #4:

Remarks to the Author:

The authors carried out a large whole-exome sequencing study of alcohol consumption using data from UK Biobank. As noted by the authors, alcohol is related to a wide range of diseases, thus investigating the genetic architecture of alcohol consumption could help future prevention and treatment for alcohol related problems. GWAS of alcohol consumption has identified hundreds of risk variant, mostly are common variants. No doubt that identifying rare coding variants could improve our understanding of the etiology of alcohol drinking behavior, with the hope that some rare variants having larger effect size than common ones.

Here are few major issues with which been addressed, this study will be improved for publication in this journal and benefit this field. I was curious why the authors didn't include all related individuals into analyses to improve power. Seems that SAIGE can handle relatedness. Given that there is a lack of GWAS in non-European samples, especially WES sample, if non-European samples can be included (the authors disclosed this as a limitation), this study will be further improved. Currently, the results are not that exciting in terms of novel findings (only one novel rare variant). Take the above two points into account might be helpful. Also, the replication is not a real replication that FinnGen is not WES data, and the checked variants are all common variants, the tested phenotypes in FinnGen is alcohol use disorder instead of alcohol consumption.

Other issues include:

- 1) Some numbers are not consistent in the Abstract/Text/Tables. For example, in the abstract, the authors claimed 34 genes, but there are only 26 genes in Table 1; 14 novel genes, but only 10 in Table 1; in line 109, 'identified one rare variant and 25 independent common...'; in line 111, '28 genes, 14 of which were novel'; in line 119, 'of the ten novel...'. Please check the numbers thoroughly.
- 2) Some issues with the citations. In line 79 and 80, the cited references were not WES study. The authors cited ref #15 in line 84, however, ref #15 is not a study of alcohol consumption. Ref #16, #17, #19, #21, #46 and #47 were repeated with previous citations, need to update the whole reference list.
- 3) Analyses. In line 108, the authors wrote 'linear regression'; suggest updating to linear mixed model if SAIGE was applied.

I was curious why rs1229984 in ADH1B, the well-known coding variant, was not in Table 1. However, it was presented in Supplementary Table 8, and it was incorrectly mapped to ADH1C and ADH1A, please clarify.

If rs1229984 has been tested, it will be the most significant association with DPW, conditional analyses on rs1229984 are needed and some results/conclusion in this locus will need to be adjusted accordingly.

Given that there is only one novel rare variant been identified, all the rest findings are common variants, I was curious have the authors looked into the associated variants (or perform conditional analysis) from DPW GWAS (PMID: 36477530) to check the novelty.

The analysis of AUDIT is loosely connected to the DPW. Curious if it is necessary to include. Not clear how many participants with AUDIT scores been analyzed.

For the gene-based results, the authors also looked up in FinnGen. Not clear the rationale that check single variant association (FinnGen) for genes identified in this study; they were from completely different analyses.

4) Presentation of the results need to be adjusted. For example, the variants in Table 1 was not well sorted: most were sorted by chr:pos, but chr11:47410415 and chr4:99347033 were listed in the last. Multiple genes were mapped for one variant, however, not clear how this was annotated (e.g., chr16:30082458 mapped to PPP4C and TBX6, instead of PPP4C only; by looking at Supplementary Table 3, I see the authors included annotation of downstream_gene_variant, not sure if this is necessary).

Table 1 presented the p-value of lookup in FinnGen, however, two AUD definitions were searched and the smallest p-value was chosen, need to clarify this.

Supplementary Table 1 contains both the variables for the primary (DPW) and secondary (AUDIT) study trait and variables for PheWAS, suggest separating the variables for PheWAS into a new table. In Figure 1, the N is 454,787. Would be better to add another N (365,195) which is the actual sample size for the primary analysis.

In Table 2, some genes and sets have the same values of results, indicating that the MAF filtering didn't generate different set of variants, please take this into account to make the table more concise (currently information was duplicated). Would be helpful if the authors can add a column of 'number of variants in analyzed set', this might answer the above question. Not clear the rationale of the leave-one-variant-out analysis.

Most of the information in Table 2 and Supplementary Table 6 were duplicated, can merge the most important information and remove one, same as Table S9 and S10.

Some statements need to be updated:

Line 66, 'previous GWAS utilized genotyping microarrays', would be better to add that those genotype data was imputed.

Supplementary Table 2, 'SE' should be 'SD'.

Line 145-146, 'previously published', do the authors mean 'previously reported'?

Cannot understand: 'the proportion of 150 participants (2.03%, 8825 LOF) with ADH1C variants or GIGYF1 (1.75%, 155 LOF and 7449 151 missense) variants were relatively high'.

Line 177, 'suggesting that coding variants do not play a large role in these GWAS regions'. This is overstated, which could be simply a power issue of the WES.

Line 327, the authors cite a conclusion from a previous study 'not marked heterogeneity across diverse ancestries', this is true for genome-wide genetic architecture, but this does not apply to single variant associations in other populations which might have population-specific causal variants.

Line 372, 'Participants who endorsed never currently drinking alcohol were included'. Should those participants be removed?

Please use 'sex' instead of 'gender' consistently.

The authors sincerely appreciate the critical reviews of the paper, and for the helpful way in which the reviewing editors put together a constructive list of suggestions for the revision of the paper. We have now revised the paper to carefully address all the points raised. Our responses below are preceded by “ - ”, and changes made to the paper are shown below within ‘...’, and in red font in the revised paper.

Detailed Responses to Reviewers

Reviewer #1 (Remarks to the Author):

This manuscript presents the results of an exome-wide association study (ExWAS) conducted on 365,196 individuals from the UK Biobank, aimed at uncovering novel genes associated with alcohol drinking per week (DPW). This study identified 34 genes, of which 16 are novel. Among these, 14 were also associated with alcohol use disorder in FinnGen. Notably, rare predicted loss-of-function variants in two novel genes, *GIGYF1* and *ANKRD12*, collectively indicated detrimental effects on DPW. Overall, this is an engaging paper with significant results.

Response:

- Thanks for your appreciation.

However, I have noted the following points for improvement:

The manuscript has multiple typographical and grammatical errors. Please correct "Notably," in line 46 and eliminate the redundant sentence in lines 230 and 231. Additionally, review the sentences around lines 286-287 and 310-311 for clarity. I couldn't pinpoint all sentences with potential issues, but a thorough review is necessary.

Response:

- According to your suggestions, we have thoroughly reviewed the manuscript for clarity and examined the typographical and grammatical errors. Meanwhile, the revised sentences around lines 290-293 and 314-319 are provided below for your convenience.
- Lines 290-293, ‘**Meanwhile**, results of the PheWAS showed significant associations of **the alcohol consumption gene**, *GIGYF1*, with blood levels of

HDL cholesterol and several inflammatory biomarkers. *Therefore, GIGYF1* may participate in the *metabolic disturbance* caused by alcohol consumption.’ Lines 314-319, ‘*Consistently with previous findings, our* PheWAS analyses *indicated* strong correlations between these genes *and* cognitive decline *as well as* altered white matter integrity, *which suggests that these genes play a significant role in brain function and structure.* The findings are plausible as *prior* studies have observed the associations between heavy alcohol consumption and *changes in brain structure.*’

In line 126, it would be beneficial to provide more context or explanation for the phrase "As expected."

Response:

- We have deleted this phrase as we think it is redundant.

On line 140, the phrase ", which was the highest among the different groups," is unclear. Please provide more context or clarification.

Response:

- We have revised this sentence as suggested.
- Lines 137-138, ‘We noted that the gene-wise burden of rare missense variants *and ultra-rare LOF variants each individually contribute to 0.29% of the phenotypic variance.*’

In line 184, instead of referring to them as "the two novel genes," please specify the names of the genes for clarity.

Response:

- We have specified the names of the two genes.
- Lines 182-184, ‘Conversely, the two novel genes *GIGYF1* and *ANKRD12* were observed to be significantly associated with alcohol consumption in both males and females’

The first sentence of the paragraph at line 225 is unclear. Please revise for better understanding.

Response:

- We have revised the sentences and made it clear to be understood.

- Lines 225-226, 'Alcohol consumption has been reported to be associated with a wide range of biological markers such as metabolites and health outcomes.'

On line 264, I suggest adding "but" before "our study" to improve the flow and coherence of the sentence.

Response:

- We have added "but" before "our study".

In Figure 2 (A) and (B), the differentiation between circles and squares is not clear. Please consider improving this for better comprehension.

Response:

- According to your suggestions, we have replaced the squares with triangles to enhance comprehension.

Reviewer #2 (Remarks to the Author):

In this study, the authors undertake a GWAS and ExWAS analysis of alcohol use using several metrics. Their approach is sound and the methods are valid.

This is the first ExWAS report of for this trait in UKB that I am aware of, so it has some novelty. Of the highlighted "novel" genes, ANKRD12 and GIGYF1 are well known genes for reduced cognitive function. GIGYF1 has a multitude of additional associations, as mentioned in this manuscript. Please see references below suggesting these are primarily cognitive dysfunction / intellectual disability genes.

My major concern is the idea that these are genes primarily related to alcohol consumption in some way, or that some of the PheWAS associations may be mediated by alcohol use disorder. I would recommend the authors leverage MR to understand whether cognitive function / income / Townsend deprivation index (all available in UKB) alterations mediate reduced alcohol use (likely) or vice versa (would be surprising). This could be done with both the rare variants of interest (in GIGYF1 and ANKRD12) and common variants more broadly related to these traits.

Refs:

ANKRD12 & cognition:

<https://www.nature.com/articles/s41588-023-01398-8>

GIGYF1 and cognition:

<https://www.nature.com/articles/s41588-023-01398-8>

<https://www.nature.com/articles/s41588-022-01104-0/figures/3>

<https://www.deciphergenomics.org/gene/GIGYF1/overview/clinical-info>

Response:

- We appreciate your thoughtful insights and acknowledge the significance of investigating the potential mediating relationships between *ANKRD12*, *GIGYF1*, cognition, and alcohol consumption. In response to your valuable suggestion, we have diligently conducted this analysis and have incorporated the findings into both the results and discussion sections of our manuscript. As you suggested, we first conducted a bidirectional Mendelian randomization (MR) analysis between cognition and alcohol consumption. We employed GWAS summary data for the general factor of intelligence, derived from a compilation of seven distinct cognitive tests ¹, all sourced from the UK Biobank. Ensuring the avoidance of sample overlap, we utilized separate GWAS summary data for alcohol consumption, excluding participants from the UK Biobank ². However, our MR analysis did not reveal any causal relationships between cognition and alcohol consumption in either direction. Specifically, the findings indicated no significant effect of alcohol consumption on cognition ($\beta = -0.04$, 95%CI = -0.19-0.11, $P = 0.61$), nor did cognition exert a substantial impact on alcohol consumption ($\beta = 0.003$, 95%CI = -0.11-0.12, $P = 0.96$). This result aligns consistently with prior research ³. Given the limited evidence substantiating causal links between cognition and alcohol consumption, it is plausible that the associations observed with cognition and alcohol consumption may stem from the pleiotropic effects of *ANKRD12* and *GIGYF1*. We have added the analysis to the Results and Supplementary Results sections.
- Lines 249-252, ‘Given the limited evidence supporting causal links between cognition and alcohol consumption (see Supplementary Information for

details), it is plausible that the observed associations may stem from the pleiotropic effects of *ANKRD12* and *GIGYF1*.’

- Moreover, we have added more discussion about the influence of these two novel genes on cognitive function and intellectual disability.
- Lines 313-317, ‘Moreover, *ANKRD12* and *GIGYF1* are well-known genes for reduced cognitive function and intellectual disability as evidenced by previous studies^{4,5}. Consistently with previous findings, our PheWAS analyses indicated strong correlations between these genes and cognitive decline as well as altered white matter integrity, which suggests that these genes play a significant role in brain function and structure.’

Reference:

1. de la Fuente J, Davies G, Grotzinger AD, Tucker-Drob EM, Deary IJ. A general dimension of genetic sharing across diverse cognitive traits inferred from molecular data. *Nat Hum Behav.* 2021;5(1):49-58. doi:10.1038/s41562-020-00936-2
2. Liu M, Jiang Y, Wedow R, et al. Association studies of up to 1.2 million individuals yield new insights into the genetic etiology of tobacco and alcohol use. *Nat Genet.* 2019;51(2):237-244. doi:10.1038/s41588-018-0307-5
3. Mahedy L, Suddell S, Skirrow C, et al. Alcohol use and cognitive functioning in young adults: improving causal inference. *Addiction.* 2021;116(2):292-302. doi:10.1111/add.15100
4. Fu, J.M. et al. Rare coding variation provides insight into the genetic architecture and phenotypic context of autism. *Nat Genet* 54, 1320-1331 (2022).
5. Chen, C.Y. et al. The impact of rare protein coding genetic variation on adult cognitive function. *Nat Genet* 55, 927-938 (2023).

Otherwise, I think the study is reasonably well executed, and other than some minor grammatical errors or typos, reasonably well written.

Response:

- We have thoroughly reviewed the manuscript and revised the typographical and grammatical errors.

Reviewer #3 (Remarks to the Author):

This study performed whole exome sequencing analysis of alcohol consumption in UKB. As the authors point out, most previous studies have focused on common variants.

Overall this study is of high interest and relevance, and offers novel insights into the mechanisms by which alcohol impacts the body. The authors found 16 novel genes associated with alcohol, including 2 novel LOF genes. They also performed several additional analyses, including cell enrichment, PhWAS which adds to the story. Particularly interesting are associations with genes linked to alcohol metabolism (ADH1C), enrichment in brain regions impacted by alcohol e.g. cerebellum, and sex differences.

Specific comments below. As I do not have specific expertise on rare variant analyses, I have focused the critique to areas within my knowledge:

1. Abstract -

i)"disturbance of white matter integrity" - not sure this can be inferred given the measures are cross-sectional.

Response:

- Thanks for the suggestions. We have revised the description.
- Lines 50-52, 'Phenome-wide association analyses revealed that the loss-of-function variants in alcohol consumption genes were associated with brain white matter integrity and risk of cardiovascular, digestive, and neuropsychiatric diseases.'

ii) "..showed deleterious effects on DPW" - I was not clear which direction this meant.

Response:

- We have revised the description as suggested.
- Lines 45-47, 'Rare predicted loss-of-function variants in the two novel genes *GIGYF1* and *ANKRD12* in aggregate were correlated with lower level of alcohol consumption ($\beta_{\text{burden}}=-0.0126$ to -0.0206).'

2. Methods-

i) How were European ancestry participants identified? It appears to be self-reported ("estimated ancestry") - surely using genetic ancestry would be more robust?

Response:

- We have included the description of the ancestry estimation in the Supplementary Methods.
- Supplementary Methods: Lines 115-118, 'White British individuals were

identified as the intersection of participants who self-reported as 'White British' and those who exhibited very similar genetic ancestry in the principal component analysis of genotypes. Detailed information about quality control for population structure can be found at <https://biobank.ndph.ox.ac.uk/showcase/refer.cgi?id=155580>.'

ii) I think giving a rationale for use of different alcohol phenotypes e.g. AUDIT measures is important.

Response:

- We appreciate your comments. We have provided the rationale for using AUDIT measures in the Supplementary Methods.
- Supplementary Methods: Lines129-135, 'In order to comprehensively assess various aspects of alcohol-related behaviors, we employed a range of alcohol phenotypes, including measurements derived from the Alcohol Use Disorders Identification Test (AUDIT) ¹. AUDIT, a comprehensive assessment tool, encompasses parameters such as alcohol consumption, dependency, and risky patterns, among others. Importantly, AUDIT measures have the capacity to identify individuals at risk of alcohol use disorders. In an effort to extend the implications of alcohol consumption findings to alcohol use disorders, we conducted ExWAS of AUDIT measures. In an effort to extend the implications of alcohol consumption findings to alcohol use disorders, we conducted ExWAS of AUDIT measures.'

Reference:

1. Saunders, J. B., Aasland, O. G., Babor, T. F., de la Fuente, J. R. & Grant, M. Development of the Alcohol Use Disorders Identification Test (AUDIT): WHO Collaborative Project on Early Detection of Persons with Harmful Alcohol Consumption--II. *Addiction* 88, 791-804, doi:10.1111/j.1360-0443.1993.tb02093.x (1993).

iii) DPW in alcohol papers usually means drinks per week (US drinks which do not equate to UK units). Here it appears to mean log-transformed UK units? Terminology is confusing.

Response:

- Thank you for bringing up this important point regarding terminology. We have replaced the term "DPW" with "alcohol consumption" in the manuscript.

iv) Drinking up to 3 times monthly was assigned weekly volume of 0 - this does not

seem correct?

Response:

- We acknowledge your concern regarding the accuracy of assigning a weekly value of 0 to individuals who reported drinking up to 3 times per month. However, it is important to note that data on weekly alcohol consumption were not collected for individuals who reported their current intake frequency as "one to three times per month," "special occasions only," or "never." To address this limitation, we followed the previous study¹ and assumed a weekly alcohol consumption volume of 0 for these participants. We have added more details to the Methods section.
- Lines 381-383, 'Individuals reporting current intake frequency of "one to three times a month", "special occasions only" or "never" (for whom the weekly alcohol consumption was not collected), were assumed to have a weekly alcohol consumption volume of 0.'
- Furthermore, we have acknowledged this limitation in the discussion section.
- Lines 344-348, 'Lastly, participants who drinking up to 3 times monthly and less were assigned a weekly drinking level of zero following a previous study¹. While this simplified approach may introduce some error into their drinking levels, it is expected to be relatively small given the infrequency of their alcohol consumption.'

Reference

1. Howe, L.J. et al. Genetic evidence for assortative mating on alcohol consumption in the UK Biobank. Nature communications 10, 1-10 (2019).

v) What was done with former drinkers? Many are "sick quitters" so can influence results and need to be handled with care in analyses. vi) Were non-drinkers excluded?

I was not clear.

Response:

- We appreciate your comments and concur with your opinions. We retained both former drinkers and non-drinkers in our analysis to maintain sample representativeness and explore genetic factors across diverse alcohol consumption patterns. We have excluded the former drinkers and non-drinkers

from the analysis as part of a sensitivity analysis. After exclusion of former drinkers and non-drinkers, most of the identified genes in the main analysis remained robust and all the effect directions remained the same. We have added the analysis to the revised manuscript.

- Supplementary Methods: Lines 212-214, ‘Several sensitivity analyses were conducted to evaluate the robustness of our results. Initially, we excluded participants who were former drinkers and non-drinkers (Field 20117) and performed ExWAS again.’
- Supplementary Results: Lines 264-271, ‘We performed several sensitivity analyses to evaluate the robustness of our results. To evaluate the impacts of former drinkers and non-drinkers on our results, we excluded former drinkers and non-drinkers and performed ExWAS again. After exclusion of former drinkers and non-drinkers, most of the identified genes in the main analysis remained robust and all the effect directions remained the same. In the ExWAS for single variants, 17 of the identified 26 (65%) loci remained genome-wide significant ($P < 5 \times 10^{-8}$) (Supplementary Table 8). In gene-based collapsing association analysis, 12 of the identified 15 (80%) associations remained significant ($P_{\text{bonf}} < 0.05$, $P < 0.05/20096$) (Supplementary Table 12).’

vii) For the PhWAS it was not clear to me how/why the phenotypes were selected? This includes the MRI phenotypes. What was corrected for in this analyses? "214 nervous traits" I think refers to MRI phenotypes? If so then a there is probably a better term.

Response:

- Thank you for your thoughtful questions and observations. We chose phenotypes that are crucial for understanding the impact of alcohol consumption genes on human health and function or could potentially result from alcohol consumption. We have provided the rationale for the selection of phenotypes of the PheWAS in the Supplementary Methods (Lines 180-194).
- The PheWAS analyses were corrected for the covariates including age, sex, and top 10 genetic components. We have incorporated the details regarding the correction methods applied in the analyses in the Methods (Lines 509-511).
- We have modified the term "214 nervous traits " to "214 neuroimaging traits".

3. Results -

i) Average DPW = 1.98? Again, see point above, this is confusing as I originally read as drinks per week and thought it was wrong as too low. I think giving drinks/units per week is more informative for reader.

Response:

- We acknowledge the confusion caused by the term "DPW" and the potential misinterpretation. We agree with you that providing the information in terms of drinks or units per week is more informative. We have incorporated the data on alcohol units per week in the revised manuscript.
- Lines 386-387, 'The median alcoholic units per week of the whole sample was 10 (Supplementary Table 2).'

ii) What "white matter measures" in the corpus callosum were examined/associated?

Response:

- We utilized the fractional anisotropy (FA; the directional coherence of water molecule diffusion) of white matter tracts within the corpus callosum. We have included details regarding the measurements of white matter data in the Results (Line 252-256) and Supplementary Methods (Line 166-179).
- Lines 252-256, 'The variant-phenotype association analyses revealed significant correlations with the fractional anisotropy (FA) of various white matter tracts. Notably, significant correlations were observed for FA in specific regions, including genu of corpus callosum ($\beta=0.068$, $P=7.90e-14$), left superior frontal-occipital fasciculus ($\beta=0.065$, $P=1.06e-12$), and anterior limb of the internal capsule ($\beta=0.066$, $P=1.84e-12$).'

4. Discussion-

i) Grammatical errors in a couple of places, which looked out of place in an otherwise nicely written paper.

Response:

- We have thoroughly reviewed the manuscript and revised the typographical and grammatical errors.

5. Supplement -

i) It seems that the authors ran their own pre-processing MRI pipeline - what was the rationale for this when UKB imaging has been pre-processed and QC'd by team and IDPs used by most other publications? There is no detail about the white matter

measures anywhere I could see in the paper.

Response:

- We appreciate your query regarding our MRI preprocessing approach. While the UK Biobank team provided pre-processed and quality-controlled imaging data, we opted to utilize our own preprocessing pipeline for several reasons. We opted to utilize the automated anatomical labeling 3 (AAL3)¹ template due to its finer parcellation, especially in the subcortical regions. Subcortical structures play crucial roles in reward processing, emotion regulation, and decision making, which are closely linked to alcohol use and addiction. Precise subcortical divisions in the AAL3 template facilitates the identification of the neural basis underlying the association between genetic variants and alcohol-related behaviors. We have included details explaining the rationale for using the MRI pipeline and provided comprehensive information about the white matter measures in the Supplementary Methods (Lines 163-165, Lines 166-179).

ii) Cognitive tests - which timepoints were used for tests? Seems like a mixture.

Response:

- We appreciate your observation about the mixture of timepoints. In UK Biobank, majority of cognitive tests were conducted at the baseline measurement, while another subset of cognitive tests was administered during imaging follow-up sessions. We utilized cognitive tests from both the baseline and imaging follow-up. Specifically, for cognitive tests assessed in different timepoints, we selected the timepoints that corresponded to the maximum sample size for each cognitive test. We have provided detailed information about the specific timepoints for each cognitive test in the Supplementary Methods (Lines 195-210).

iii) SFigures: SFig 10-14 the text is unreadably small, SFig 19 does not spell out abbreviations, SFig 25 does not spell out abbreviations and some text unreadable as overlapping.

Response:

- We have addressed text size appropriately and incorporated abbreviation clarifications to enhance overall understanding.

Reviewer #4 (Remarks to the Author):

The authors carried out a large whole-exome sequencing study of alcohol consumption using data from UK Biobank. As noted by the authors, alcohol is related to a wide range of diseases, thus investigating the genetic architecture of alcohol consumption could help future prevention and treatment for alcohol related problems. GWAS of alcohol consumption has identified hundreds of risk variant, mostly are common variants. No doubt that identifying rare coding variants could improve our understanding of the etiology of alcohol drinking behavior, with the hope that some rare variants having larger effect size than common ones.

Here are few major issues with which been addressed, this study will be improved for publication in this journal and benefit this field. I was curious why the authors didn't include all related individuals into analyses to improve power. Seems that SAIGE can handle relatedness. Given that there is a lack of GWAS in non-European samples, especially WES sample, if non-European samples can be included (the authors disclosed this as a limitation), this study will be further improved. Currently, the results are not that exciting in terms of novel findings (only one novel rare variant). Take the above two points into account might be helpful.

Response:

- Thank you for your thoughtful comments. In our primary analysis, we excluded individuals with genetic relationships up to the 3rd degree or closer, aligning with established methodologies in previous study ¹. Following your suggestions, we have included all related participants to enhance statistical power. In the ExWAS for single variants, we identified 26 loci significantly associated with alcohol consumption, including 6 novel genes. In the gene-based collapsing association analysis, 12 associations (covering 4 genes) achieved significance after Bonferroni correction ($P < 0.05/20,096$). Additionally, we identified 19 genes with an overall false discovery rate < 0.05 , of which 14 were novel. We have incorporated these analyses into the Supplementary Methods (Lines 218-219) and Supplementary Results (Lines 280-292) sections.

- According to your suggestions, we conducted ExWAS in the non-White British population. In the ExWAS for single variants, we identified one locus significantly associated with alcohol consumption, which was consistent with findings in the White British population. However, in the gene-based collapsing association analysis, we did not observe significant associations, likely due to the smaller sample size in the non-White British population. We have included these analyses in the Supplementary Methods (Lines 219-221) and Supplementary Results (Lines 293-299) sections.

Reference

1. Jurgens, S.J. et al. Analysis of rare genetic variation underlying cardiometabolic diseases and traits among 200,000 individuals in the UK Biobank. *Nat Genet* 54, 240-250 (2022).

Also, the replication is not a real replication that FinnGen is not WES data, and the checked variants are all common variants, the tested phenotypes in FinnGen is alcohol use disorder instead of alcohol consumption.

Response:

- We agree with you that FinnGen is not an ideal cohort to perfectly replicate the results from WES data. Our choice to use FinnGen was driven by several considerations. Firstly, we were unable to identify a more suitable cohort with WES data for a complete replication. Secondly, our ExWAS encompassed common variants, thus leveraging available GWAS data for replication is acceptable. We now acknowledge that our primary focus was not framed as a replication but rather aimed at providing evidence to support our results. Our results suggest that our findings from WES data are credible, at least for the common variants. Regarding the potential discrepancies between AUD and alcohol consumption, we now included a large GWAS for alcohol consumption to validate the significant common variants from WES data.
- We now refrain from claiming that the above analysis is a replication but simply report the number of common variants that were also significant in independent GWAS data. We have also included a discussion of these limitations in the manuscript.
- Lines 119-121, ‘Eight of the ten novel variants were examined in an alcohol consumption GWAS that did not include UK Biobank participants¹, and four were significant ($P < 0.05$) (Table 1, Supplementary Table 4).’

- Lines 339-342, ‘Third, due to the unavailability of a comparable large cohort with WES data for replication, we relied on existing GWAS data for alcohol consumption and AUD to support our findings. Further whole-exome studies are needed to replicate the identified genes.’

Reference

1. Liu, M. et al. Association studies of up to 1.2 million individuals yield new insights into the genetic etiology of tobacco and alcohol use. *Nature Genetics* 51, 237-244 (2019).

Other issues include:

- 1) Some numbers are not consistent in the Abstract/Text/Tables. For example, in the abstract, the authors claimed 34 genes, but there are only 26 genes in Table 1; 14 novel genes, but only 10 in Table 1; in line 109, ‘identified one rare variant and 25 independent common...’; in line 111, ‘28 genes, 14 of which were novel’; in line 119, ‘of the ten novel...’. Please check the numbers thoroughly.

Response:

- We have reviewed the numbers thoroughly to make ensure they are correct and consistent.
- Lines 42-45, ‘The exome-wide association study (ExWAS) of alcohol consumption identified 26 loci through single variant analysis and six genes through gene-based analysis, twelve of them novel. Eight of the novel alcohol consumption genes were also associated with alcohol use disorder in FinnGen.’
- Lines 110-117, ‘We identified one rare variant and 25 independent common genomic loci associated with alcohol consumption at genome-wide significance ($P < 5 \times 10^{-8}$).... Among the 25 common genomic loci, ten were novel (*rs41288799*, *rs3214499*, *rs11096989*, *rs4975020*, *rs223389*, *rs77623289*, *rs40831*, *rs2278557*, *rs12373123*, *rs1652375*)’.
- Lines 121-122, ‘Additionally, eight out of the ten novel variants were associated with alcohol use disorder ($P_{\text{one_tail}} < 0.05$) in the FinnGen study (Table 1, Supplementary Table 5).’
- Lines 260-263, ‘We identified twelve novel genes associated with alcohol consumption as well as replicated several previously reported genes, which may shed new light on pathophysiological processes in alcohol use. Eight of the novel alcohol consumption genes were also associated with alcohol use disorder in FinnGen.’

2) Some issues with the citations. In line 79 and 80, the cited references were not WES study. The authors cited ref #15 in line 84, however, ref #15 is not a study of alcohol consumption. Ref #16, #17, #19, #21, #46 and #47 were repeated with previous citations, need to update the whole reference list.

Response:

- We have examined our references thoroughly and revised the incorrect ones.
- Lines 80-85, ‘WES would greatly facilitate our understanding of the genetic architecture of alcohol consumption as well as its implication on physical and mental health¹. However, to the best of our knowledge, there have been few large-scale WES studies on alcohol consumption, let alone elucidating the potential implications of the identified genes^{2,3}. Meanwhile, as indicated by a previous genome-wide association study, significant genetic associations existed between alcohol consumption and several body health phenotypes⁴.’

Reference

1. Saunders, G.R.B. *et al.* Genetic diversity fuels gene discovery for tobacco and alcohol use. *Nature* **612**, 720-724 (2022).
2. Brazel, D.M. *et al.* Exome Chip Meta-analysis Fine Maps Causal Variants and Elucidates the Genetic Architecture of Rare Coding Variants in Smoking and Alcohol Use. *Biol Psychiatry* **85**, 946-955 (2019).
3. Marees, A.T. *et al.* Exploring the role of low-frequency and rare exonic variants in alcohol and tobacco use. *Drug Alcohol Depend* **188**, 94-101 (2018).
4. Kranzler, H.R. *et al.* Genome-wide association study of alcohol consumption and use disorder in 274,424 individuals from multiple populations. *Nat Commun* **10**, 1499 (2019).

3) Analyses. In line 108, the authors wrote ‘linear regression’; suggest updating to linear mixed model if SAIGE was applied.

Response:

- We have revised the manuscript as suggested.

I was curious why rs1229984 in ADH1B, the well-known coding variant, was not in Table 1. However, it was presented in Supplementary Table 8, and it was incorrectly mapped to ADH1C and ADH1A, please clarify.

Response:

- In the quality control step, rs1229984 was removed due to Hardy-Weinberg

equilibrium, resulting in no association result for this variant. To strengthen the validity of our gene-based analysis results, we queried the genes in FinnGen and examined the associations between the variants mapped to the genes and alcohol use disorder. In FinnGen, rs1229984 was mapped to the ADH1C and ADH1A gene, leading to the association results in Supplementary Table 8. According to your comments, we have removed this result from the revised manuscript.

If rs1229984 has been tested, it will be the most significant association with DPW, conditional analyses on rs1229984 are needed and some results/conclusion in this locus will need to be adjusted accordingly.

Response:

- In response to your comments, we performed ExWAS conditioned on the *rs1229984* and most of the associations remained robust. In ExWAS for single variants, 24 out of 26 (92.3%) loci had the same direction and 19/26 (73.1%) loci remained significant ($P < 5 \times 10^{-8}$). In gene-based collapsing association analysis, 13 out of 15 (86.7%) associations had the same direction. Notably, *GIGYF1* and *ANKRD12* remained significant (*GIGYF1*: $P = 1.17 \times 10^{-10}$, *ANKRD12*: $P = 5.81 \times 10^{-10}$) and the association for *ADH5* and *HECTD4* remained nominal significant (*ADH5*: $P = 0.02$, *HECTD4*: $P = 1.6 \times 10^{-5}$). We have included this analysis to the Supplementary Methods (Lines 214-215) and Supplementary Results (Lines 272-279).

Given that there is only one novel rare variant been identified, all the rest findings are common variants, I was curious have the authors looked into the associated variants (or perform conditional analysis) from DPW GWAS (PMID: 36477530) to check the novelty.

Response:

- We have carefully compared the common variants discovered in our analysis with the DPW GWAS (PMID: 36477530) and ten loci are novel. The novel loci have been labeled in Table 1.

The analysis of AUDIT is loosely connected to the DPW. Curious if it is necessary to include. Not clear how many participants with AUDIT scores been analyzed.

Response:

- AUDIT is a comprehensive assessment tool that evaluates alcohol consumption, dependency, and risky patterns. Notably, AUDIT measures have the potential to identify individuals at risk of alcohol use disorders. We utilized the AUDIT measures to validate whether the conclusions drawn from alcohol consumption findings could be extrapolated to alcohol use disorders. This may strengthen the robustness and clinical significance of our conclusions. However, we acknowledge that the results of AUDIT were not highly significant. This could be attributed to the limited sample size, resulting in insufficient statistical power. We have relocated this portion of the results to the Supplementary Results (Lines 231-246).
- The sample size is 118,666. We have included a detailed description in the Supplementary Methods (Lines 142-144).

For the gene-based results, the authors also looked up in FinnGen. Not clear the rationale that check single variant association (FinnGen) for genes identified in this study; they were from completely different analyses.

Response:

- We appreciate your insightful comment. Our decision to utilize the results of single variant association analysis in FinnGen to support our results of our gene-based analysis was driven by several considerations. Firstly, finding a large dataset with identical analysis methods for direct validation was challenging. Secondly, our primary goal was not direct validation but gathering additional evidence for our gene-level associations. While these analyses differ, they can complement each other, providing varying levels of evidence for the involvement of specific genes in alcohol consumption. We acknowledge the limitations of this approach and, in response to your concerns, have removed this analysis from the revised manuscript.

4) Presentation of the results need to be adjusted. For example, the variants in Table 1 was not well sorted: most were sorted by chr:pos, but chr11:47410415 and chr4:99347033 were listed in the last. Multiple genes were mapped for one variant, however, not clear how this was annotated (e.g., chr16:30082458 mapped to PPP4C and TBX6, instead of PPP4C only; by looking at Supplementary Table 3, I see the

authors included annotation of downstream_gene_variant, not sure if this is necessary).

Response:

- We have sorted the variants in **Table 1** and performed the annotation using Annovar¹. The results indicate that most of the variants are mapped to a single gene, except for *rs41288799* (mapped to *PREB* and *ABHD1*) and *rs1652375* (mapped to *RMCI* and *NPCI*).

Reference

1. Wang K, Li M, Hakonarson H. ANNOVAR: functional annotation of genetic variants from high-throughput sequencing data. *Nucleic Acids Res.* 2010 Sep;38(16):e164. doi: 10.1093/nar/gkq603IF: 14.9 Q1 . Epub 2010 Jul 3. PMID: 20601685; PMCID: PMC2938201.

Table 1 presented the p-value of lookup in FinnGen, however, two AUD definitions were searched and the smallest p-value was chosen, need to clarify this.

Response:

- To avoid confusion, we have presented the results of both AUD definitions in Table 1.

Supplementary Table 1 contains both the variables for the primary (DPW) and secondary (AUDIT) study trait and variables for PheWAS, suggest separating the variables for PheWAS into a new table.

Response:

- We have separated the variables for PheWAS into the revised Supplementary Table 22 as suggested.

In Figure 1, the N is 454,787. Would be better to add another N (365,195) which is the actual sample size for the primary analysis.

Response:

- We have added the actual sample size (365,195) for the primary analysis to the Figure 1.

In Table 2, some genes and sets have the same values of results, indicating that the MAF filtering didn't generate different set of variants, please take this into account to make the table more concise (currently information was duplicated). Would be helpful if the authors can add a column of 'number of variants in analyzed set', this might answer the above question.

Response:

- We have added the information on the number of variants in analyzed set as suggested.

Not clear the rationale of the leave-one-variant-out analysis.

Response:

- Your inquiry regarding the rationale for the leave-one-variant-out analysis is appreciated. This analysis involved iteratively excluding individual variants from the corresponding gene-based collapsing group. This was undertaken to address specific aspects: firstly, to assess the robustness and consistency of the results across variant exclusions; secondly, to discern whether the gene-based collapsing association results were predominantly driven by specific variants; and finally, to investigate whether the observed gene-based collapsing associations were influenced by numerous rare variants characterized by relatively small effect sizes. This analytical approach allows for a comprehensive evaluation of the role of individual variants within the broader gene-based context. We have provided the rationale of the leave-one-variant-out analysis in the Supplementary Methods (Lines 222-230).

Most of the information in Table 2 and Supplementary Table 6 were duplicated, can merge the most important information and remove one, same as Table S9 and S10.

Response:

- We have revised the manuscript as suggested.

Some statements need to be updated:

Line 66, 'previous GWAS utilized genotyping microarrays', would be better to add that those genotype data was imputed.

Response:

- We have revised the manuscript as suggested.

Supplementary Table 2, 'SE' should be 'SD'.

Response:

- We have revised the manuscript as suggested.

Line 145-146, 'previously published', do the authors mean 'previously reported'?

Response:

- We have replaced 'previously published' with 'previously reported'.

Cannot understand: ‘the proportion of 150 participants (2.03%, 8825 LOF) with ADH1C variants or GIGYF1 (1.75%, 155 LOF and 7449 151 missense) variants were relatively high’.

Response:

- We have revised the sentences and made it clear to be understood.
- Lines 146-147 ‘2.03% (n = 8825) of the participants carried a LOF variant located in ADH1C exons and GIGYF1 variants were carried by 1.75% (n = 7604) participants (Fig.3D).’

Line 177, ‘suggesting that coding variants do not play a large role in these GWAS regions’. This is overstated, which could be simply a power issue of the WES.

Response:

- We have revised the manuscript as suggested.
- Lines 174-176, ‘The influence of coding variants within the GWAS regions did not show substantial effects. This could potentially be attributed to the limited statistical power of ExWAS.’

Line 327, the authors cite a conclusion from a previous study ‘not marked heterogeneity across diverse ancestries’, this is true for genome-wide genetic architecture, but this does not apply to single variant associations in other populations which might have population-specific causal variants.

Response:

- We appreciate your comments and concur with your opinions. We have added this in the limitation part.
- Lines 335-337, ‘Although a previous GWAS of alcohol use indicated no marked heterogeneity across diverse ancestries¹, single variant association analysis in other populations might identify population-specific variants.’

Reference

1. Saunders, G.R.B. et al. Genetic diversity fuels gene discovery for tobacco and alcohol use. Nature 612, 720-724 (2022).

Line 372, ‘Participants who endorsed never currently drinking alcohol were included’.

Should those participants be removed?

Response:

- We appreciate your comments. We retained both former drinkers and non-drinkers in our analysis to maintain sample representativeness and explore genetic factors across diverse alcohol consumption patterns. We have excluded the former drinkers and non-drinkers from the analysis as part of a sensitivity analysis. After exclusion of former drinkers and non-drinkers, most of the identified genes in the main analysis remained robust and all the effect directions remained the same. We have added the analysis to the revised manuscript.
- Supplementary Methods: Lines 212-214, ‘Several sensitivity analyses were conducted to evaluate the robustness of our results. Initially, we excluded participants who were former drinkers and non-drinkers (Field 20117) and performed ExWAS again.’
- Supplementary Results: Lines 264-271, ‘We performed several sensitivity analyses to evaluate the robustness of our results. To evaluate the impacts of former drinkers and non-drinkers on our results, we excluded former drinkers and non-drinkers and performed ExWAS again. After exclusion of former drinkers and non-drinkers, most of the identified genes in the main analysis remained robust and all the effect directions remained the same. In the ExWAS for single variants, 17 of the identified 26 (65%) loci remained genome-wide significant ($P < 5 \times 10^{-8}$) (**Supplementary Table 8**). In gene-based collapsing association analysis, 12 of the identified 15 (80%) associations remained significant ($P_{\text{bonf}} < 0.05$, $P < 0.05/20096$) (**Supplementary Table 12**).’

Please use ‘sex’ instead of ‘gender’ consistently.

Response:

- We have used ‘sex’ in the manuscript as suggested.

Reviewers' Comments:

Reviewer #1:

Remarks to the Author:

This reviewer does not have any additional comments about this paper.

Reviewer #2:

Remarks to the Author:

The authors have addressed my major concerns.

Reviewer #3:

Remarks to the Author:

The authors have done a thorough job of rebutting and addressing all my previous comments. I have no further concerns.

Reviewer #4:

Remarks to the Author:

Thank the authors for addressing the questions and conducting additional analyses. Now this manuscript has been improved and strengthened.

Here are a few minor questions that I believe are trivial.

- 1). The authors mentioned "an alcohol consumption GWAS that did not include UK Biobank participants". According to my knowledge, the publicly available GWAS summary statistics from GSCAN study (Liu et al; Saunders et al) did not include 23andMe. Please double-check this.
- 2). For p-values that are barely significant ($p < 0.05$), I would suggest using 'nominally significant' instead of 'significant'. Sometimes, multiple testing correction should be considered, even for lookup/replication.
- 3). In Table 2, the authors added information as suggested. Now I understand that for some groups in the same gene, basically the same set of variants were tested. For example, GIGYF1-lof-0.0001 has identical numbers of rare and ultra-rare variants as GIGYF1-lof-0.001 (and GIGYF1-lof-0.01). This means these 3 groups tested the exact same thing. I was thinking, how to present this kind of 'duplicated' results? No need to reply, but can think about how to improve the presentation.
- 4). Would be helpful to have more description for each Supplementary table besides the title.
- 5). This is no title in Supplementary Table 3, although it was mentioned in the 'Table of Contents'.

I don't have further questions.

The authors sincerely appreciate the critical reviews of the paper, and for the helpful way in which the reviewing editors put together a constructive list of suggestions for the revision of the paper. We have now revised the paper to carefully address all the points raised. Our responses below are preceded by “ - ”, and changes made to the paper are shown below within ‘ ... ’, and in red font in the revised paper.

Detailed Responses to Reviewers

Reviewer #1 (Remarks to the Author):

This reviewer does not have any additional comments about this paper.

Reviewer #2 (Remarks to the Author):

The authors have addressed my major concerns.

Reviewer #3 (Remarks to the Author):

The authors have done a thorough job of rebutting and addressing all my previous comments. I have no further concerns.

Reviewer #4 (Remarks to the Author):

Thank the authors for addressing the questions and conducting additional analyses. Now this manuscript has been improved and strengthened.

Here are a few minor questions that I believe are trivial.

1). The authors mentioned “an alcohol consumption GWAS that did not include UK Biobank participants”. According to my knowledge, the publicly available GWAS summary statistics from GSCAN study (Liu et al; Saunders et al) did not include 23andMe. Please double-check this.

Response:

- We have verified the GWAS data used in our study and can confirm that it did not involve UK Biobank participants.

2). For p-values that are barely significant ($p < 0.05$), I would suggest using ‘nominally

significant' instead of 'significant'. Sometimes, multiple testing correction should be considered, even for lookup/replication.

Response:

- We have revised this sentence as suggested.
- Lines 111-113, 'Eight of the ten **unreported** variants were examined in an alcohol consumption GWAS that did not include UK Biobank participants, and four were **nominally** significant ($P < 0.05$) (Table 1, Supplementary Table 4).'

3). In Table 2, the authors added information as suggested. Now I understand that for some groups in the same gene, basically the same set of variants were tested. For example, GIGYF1-lof-0.0001 has identical numbers of rare and ultra-rare variants as GIGYF1-lof-0.001 (and GIGYF1-lof-0.01). This means these 3 groups tested the exact same thing. I was thinking, how to present this kind of 'duplicated' results? No need to reply, but can think about how to improve the presentation.

4). Would be helpful to have more description for each Supplementary table besides the title.

Response:

- We have provided detailed descriptions of the Supplementary tables below each table.

5). This is no title in Supplementary Table 3, although it was mentioned in the 'Table of Contents'.

Response:

- The title for Supplementary Table 3 is located in the left-most column of the table.

I don't have further questions.

Reviewers' Comments:

Reviewer #2:

Remarks to the Author:

The manuscript reads well and my opinion is unchanged from prior with one exception. In light of the methodological oversight related to analyzing all ancestries instead of European British by the authors in the initial version, I looked more carefully for issues of population stratification. It's unclear what the authors mean by the 10 ancestral PCs that were adjusted for in their mixed effects model. Are these derived from the common variation? The rare variation? Both? One good practice for proper control of population stratification is to account for both, using 10-20 PCs from common variation and 10-20 PCs from rare variation. I'm unclear on what the authors did here, so it would be helpful to have some additional information.

Finally, I'd like to re-emphasize the pleiotropy for GIGYF1 and ANKRD12 and the reasoning behind what causes reduced alcohol consumption in carriers of loss of function in these genes. Both are associated with cognitive function based on Chen et al., Nat Genet 2023 and GIGYF1 is an "autism genes" based on Fu et al., Nat Genet 2022. The authors would benefit from a more thorough PheWAS of the traits. There are some interesting traits on the GeneBass browser, for example:

For GIGYF1, the association with EtOH is significant and lower on the list from diabetes and LDL (as discussed by the authors), but also the association with Townsend Deprivation Index: <https://app.genebass.org/gene/ENSG00000146830?burdenSet=pLoF&phewasOpts=1&resultLayout=full>

Individuals with a higher Townsend Deprivation Index could be those who have less access to alcohol. It is also interesting that these individuals have lower handgrip strength and an increased chance of being a current smoker.

For ANKRD12, the association with EtOH is exome-wide significant and lower on the list from several other notable associations:

<https://app.genebass.org/gene/ENSG00000101745?burdenSet=pLoF&phewasOpts=1&resultLayout=full>

Here we again see Townsend Deprivation index and hand grip strength among the traits.

So I still think, despite absence of Mendelian Randomization evidence (notably affected by not having well-powered instruments in this case), the most parsimonious explanation might be that reduced cognitive function on average in these carriers leads to increased material deprivation (higher Townsend Deprivation Index) and in turn reduced access to alcoholic beverages on average.

Reviewer #4:

Remarks to the Author:

An issue was disclosed by the authors that exome-wide association analysis was performed in the entire unrelated sample, rather than limiting it solely to the European (White British) population. Now the authors added the ancestry-specific (European) analyses. Although most of the results were not changed (UK Biobank is dominated by White British participants), except for a few associations close to the threshold, the mixed-ancestry analysis is not appropriate.

Given that the authors have conducted the ancestry-specific analyses, I strongly suggest presenting the ancestry-specific results only or primarily (either remove or move the mixed-ancestry results to supplementary), the downstream results also need to be reconstructed. There is still valuable information added to the literature, I am still enthusiastic to see this study being published after re-organizing the manuscript.

In several places in the Results, need to specify the number of analyzed samples, one is the related participants, and one is the ancestry-specific analysis.

The authors sincerely appreciate the critical reviews of the paper, and for the helpful way in which the reviewing editors put together a constructive list of suggestions for the revision of the paper. We have now revised the paper to carefully address all the points raised. Our responses below are preceded by “ - ”, and changes made to the paper are shown below within ‘...’, and in red font in the revised paper.

Detailed Responses to Reviewers

Reviewer #2 (Remarks to the Author):

The manuscript reads well and my opinion is unchanged from prior with one exception. In light of the methodological oversight related to analyzing all ancestries instead of European British by the authors in the initial version, I looked more carefully for issues of population stratification. It's unclear what the authors mean by the 10 ancestral PCs that were adjusted for in their mixed effects model. Are these derived from the common variation? The rare variation? Both? One good practice for proper control of population stratification is to account for both, using 10-20 PCs from common variation and 10-20 PCs from rare variation. I'm unclear on what the authors did here, so it would be helpful to have some additional information.

Response:

- Thank you for your thorough review of our manuscript and insightful comments. We have now focused on the results from the white British sample and updated the corresponding results in the revised manuscript.
- For the derivation of the top 10 ancestral principal components, we followed methods outlined in a previous study ¹. Specifically, the top 10 ancestral principal components were calculated using a high-quality independent autosomal variants subset. This subset was selected based on the following parameters: MAF > 0.1%, HWE P > 10⁻⁶, missingness < 1%, and underwent two rounds of pruning (--indep-pairwise 200 100 0.1 and --indep-pairwise 200 100 0.05 in PLINK). Thus, the top 10 PCs were derived from both rare variants

and common variants. We have included a detailed description of the method to calculate the top 10 PCs in the Methods section (lines 398-406).

- Additionally, we calculated the genomic control lambda, which yielded a value of 1.04, indicating that our association statistics are not systematically inflated by population stratification (line 99).
- Lines 398-406: ‘To control population stratification, we generated the top 10 ancestral principal components (PCs) using a high-quality independent autosomal variants subset, as outlined in a prior study ¹. Specifically, this subset of variants comprised variants with MAF > 0.1%, HWE P > 10⁻⁶, missingness < 1%, and underwent two rounds of pruning (--indep-pairwise 200 100 0.1 and 200 100 0.05 in PLINK).’

References:

1 Jurgens SJ, Choi SH, Morrill VN, et al. Analysis of rare genetic variation underlying cardiometabolic diseases and traits among 200,000 individuals in the UK Biobank. Nat Genet. 2022;54(3):240-250. doi:10.1038/s41588-021-01011-w

Finally, I'd like to re-emphasize the pleiotropy for GIGYF1 and ANKRD12 and the reasoning behind what causes reduced alcohol consumption in carriers of loss of function in these genes. Both are associated with cognitive function based on Chen et al., Nat Genet 2023 and GIGYF1 is an "autism genes" based on Fu et al., Nat Genet 2022. The authors would benefit from a more thorough PheWAS of the traits. There are some interesting traits on the Genebase browser, for example:

For GIGYF1, the association with EtOH is significant and lower on the list from diabetes and LDL (as discussed by the authors), but also the association with Townsend Deprivation Index:

<https://app.genebase.org/gene/ENSG00000146830?burdenSet=pLoF&phewasOpts=1&resultLayout=full>

Individuals with a higher Townsend Deprivation Index could be those who have less access to alcohol. It is also interesting that these individuals have lower handgrip strength and an increased chance of being a current smoker.

For ANKRD12, the association with EtOH is exome-wide significant and lower on the list from several other notable associations:

<https://app.genebase.org/gene/ENSG00000101745?burdenSet=pLoF&phewasOpts=1&resultLayout=full>

Here we again see Townsend Deprivation index and hand grip strength among the traits. So I still think, despite absence of Mendelian Randomization evidence (notably affected by not having well-powered instruments in this case), the most parsimonious explanation might be that reduced cognitive function on average in these carriers leads to increased material deprivation (higher Townsend Deprivation Index) and in turn reduced access to alcoholic beverages on average.

Response:

- We appreciate your thorough examination and additional insights into the mechanisms underlying reduced alcohol consumption in carriers of loss-of-function variants in *GIGYF1* and *ANKRD12*. We agree that the associations between these genes and the Townsend deprivation index, as well as cognitive function, may shed light on the pathways through which these genes influence alcohol consumption. We have incorporated this discussion into the revised manuscript.
- Lines 299-303 ‘Interestingly, *ANKRD12* and *GIGYF1* are associated with a higher Townsend deprivation index, which could possibly lead to a less access to alcohol ¹. Given that those genes were associated with cognitive function in our PheWAS results and in previous studies ^{2,3}, it is possible that reduced cognitive function in the gene carriers results in increased material deprivation and in turn reduced alcohol consumption.’

References:

- 1 Karczewski, K.J. et al. Systematic single-variant and gene-based association testing of thousands of phenotypes in 394,841 UK Biobank exomes. *Cell Genom* 2, 100168 (2022).
- 2 Fu, J.M. et al. Rare coding variation provides insight into the genetic architecture and phenotypic context of autism. *Nat Genet* 54, 1320-1331 (2022).
- 3 Chen, C.Y. et al. The impact of rare protein coding genetic variation on adult cognitive function.

Reviewer #4 (Remarks to the Author):

An issue was disclosed by the authors that exome-wide association analysis was performed in the entire unrelated sample, rather than limiting it solely to the European (White British) population. Now the authors added the ancestry-specific (European) analyses. Although most of the results were not changed (UK Biobank is dominated by White British participants), except for a few associations close to the threshold, the mixed-ancestry analysis is not appropriate.

Given that the authors have conducted the ancestry-specific analyses, I strongly suggest presenting the ancestry-specific results only or primarily (either remove or move the mixed-ancestry results to supplementary), the downstream results also need to be reconstructed. There is still valuable information added to the literature, I am still enthusiastic to see this study being published after re-organizing the manuscript.

Response:

- Thank you for your insightful comments and valuable suggestions regarding our manuscript. We appreciate your thorough review and acknowledge the importance of addressing population stratification issues in genetic association studies.
- In response, we have carefully reconducted all the analyses using the white British sample and have removed the mixed-ancestry results to enhance the fluency of the manuscript. While most of the results remain consistent with our previous findings, we have also presented some additional findings that are specific to the white British population. In the single-variant association analysis, we have now identified 2 rare variants and 23 independent common variants. Additionally, in the gene-based association analysis, we have identified 19 associations (covering seven genes) after Bonferroni correction ($P < 0.05/19852$) and additional six putative alcohol consumption-related genes at an overall false discovery rate ($FDR < 0.05$ ($P < 1.69 \times 10^{-5}$)).

- Furthermore, we have reconstructed all downstream analyses using the white British sample. Despite these adjustments, most of the findings remained consistent with our previous analyses.
- We have updated the manuscript accordingly and believe that these revisions strengthen the quality and relevance of our study.

In several places in the Results, need to specify the number of analyzed samples, one is the related participants, and one is the ancestry-specific analysis.

Response:

- We have now specified the number of analyzed samples of the related participants and non-white participants in the revised manuscript.
- Lines 244-252: ‘Since SAIGE can handle **sample** relatedness in the regression model, we included all **373,152 white British** participants (**including both unrelated and related participants**) in the analyses to **increase** statistical power. In the ExWAS for single variants, we identified **26 independent significant variants** associated with alcohol consumption, including **four variants not detected in the unrelated white British sample, of which two were** not previously linked to alcohol consumption (Supplementary Data 26). The gene-based collapsing analysis identified **23** potential alcohol consumption-related genes with an overall FDR < 0.05. Of the 23 genes, **13** were not found in **the unrelated white British participants**, and among these, **eight** were not previously associated with alcohol consumption (Supplementary Data 27).’
- Lines 253-257: ‘Moreover, ExWAS was conducted in **61,076 unrelated** non-white **British** participants. While the ExWAS for single variants identified one locus significantly linked to alcohol consumption (Supplementary Data **28**), the gene-based collapsing analysis did not uncover any significant associations **after FDR correction**, potentially attributed to the constrained sample size among non-white **British** participants.’